# MASTER: Enhancing Large Language Model via Multi-Agent Simulated Teaching

**Liang Yue**[1*], **Yihong Tang**[1,2*], **Kehai Chen**[1†], **Jie Liu**[3], **Min Zhang**[1]
[1]Harbin Institute of Technology, Shenzhen, China
[2]Shenzhen Loop Area Institute (SLAI), Shenzhen, China
[3]Harbin Institute of Technology, China
{220110921@stu.hit.edu.cn, chenkehai@hit.edu.cn}

## Abstract

Instruction fine-tuning is crucial in NLP tasks, enhancing pretrained models' instruction-following capabilities and task-specific performance. However, obtaining high-quality fine-tuning data for large models is challenging due to data collection difficulties and high production costs. To address this, we propose MASTER, a novel data augmentation method that enriches original data through interactions among multiple agents with varying cognitive levels. We simulate three pedagogically grounded teaching scenarios, leveraging multi-agent conversations to generate high-quality teacher-student interaction data. Utilizing MASTER, we construct BOOST-QA, a fine-tuning dataset augmented from existing datasets like Orca-Math-200k, ProcQA, and OpenHermes2.5. Experiments show that models fine-tuned with BOOST-QA perform excellently across multiple benchmarks, demonstrating strong multitask generalization. Notably, MASTER significantly improves models' reasoning abilities in complex tasks, providing valuable insights for future research. Our code is publicly available at https://github.com/Toyhom/MASTER.

## 1 Introduction

In recent years, instruction-tuning or post-training has become one of the cornerstones of large language models (LLMs) [1, 2]. To meet the growing demand for data, data synthesis has been widely studied. For example, Yoo et al. combined subsets of training examples and embedded them into model prompts to generate new, high-quality textual instances [3], demonstrating notable performance improvements in text classification tasks. Ding et al. and Xu et al. proposed self-chat approaches based on predefined prompt templates and dialogue seeds, using large language models (LLMs) to generate diverse instruction data through self-dialogue [4, 5]. For example, Yoo et al. combined subsets of training examples and embedded them into model prompts to generate new, high-quality textual instances [3], demonstrating notable performance improvements in text classification tasks. While these methods enhance the diversity of synthetic instruction data, they rely heavily on manually crafted prompts and dialogue seeds and often lack clear interaction mechanisms, ultimately leading to a mismatch between the generated data and real-world instruction scenarios.

To address these challenges, our study introduces a novel multi-agent interaction framework aimed at enhancing original problem-solving datasets within simulated educational scenarios. Specifically, we developed a system comprising teacher and student agents. Through their collaborative interactions, any existing instruction dataset can be transformed into a **M**ulti-**A**gent **S**imulated **T**eaching **E**nhanced

---

[*]Equal contribution.
[†]Corresponding author.

39th Conference on Neural Information Processing Systems (NeurIPS 2025).

Resource (MASTER) framework. MASTER simulates three educational scenarios—error correction, collaborative debate, and analogical reasoning—by leveraging distinct conversational protocols and model prompts, ultimately resulting in the creation of a high-quality instruction dataset named BOOST-QA (Behaviorally Oriented Overlay of Simulated Teaching for QA).

Using the large-scale, high-quality instruction dataset BOOST-QA generated through the MASTER framework, we fine-tuned several mainstream base models. To rigorously assess the effectiveness of the MASTER method, we conducted comprehensive experiments comparing the performance of base models fine-tuned on the original datasets, datasets augmented by other methods, and those created through MASTER. The results show that BOOST-QA significantly enhances the diverse capabilities of large language models (LLMs), outperforming several existing approaches focused on data augmentation and selection.

Our main contributions are summarized as follows:

- We introduce the application of multi-agent simulated instructional scenarios in post-training data synthesis and propose a novel data augmentation method, MASTER.

- By applying MASTER to portions of Orca-math-200k, ProcQA, and OpenHermes2.5, we construct an efficient instruction fine-tuning dataset, BOOST-QA.

- We design comprehensive experiments to assess our MASTER method. Notably, in benchmark tests, multiple models fine-tuned with 19K instruction-response pairs from our BOOST-QA exhibit significant performance improvements across various task domains.

## 2 Relate Work

### 2.1 Data Synthesis and Augmentation

In recent years, data synthesis and augmentation techniques have become essential for enhancing LLM performance and generalization capabilities [6, 7]. Early approaches focused primarily on simple lexical and positional transformations or employed LLMs to generate new training samples through synonym substitution of sentences from original datasets [8–10]. While these methods partially addressed data scarcity, they risked introducing noise-induced semantic drift and often produced insufficient sample diversity for complex tasks [11]. To overcome these limitations, prompt-guided LLM approaches for comprehensive data expansion have emerged as a promising alternative, including methods such as constraint-augmented problem evolution to deepen original questions [12], multi-task contextual generation by sampling from seed pools [13], and knowledge-tree recursive QA to extend initial keywords [14]. These techniques improve content diversity and generalizability of synthetic instructions, yet remain overly reliant on predefined prompts and keywords while lacking authentic natural language contexts. In contrast to prior work, our approach enhances problem-solving data quality by simulating real classroom learning scenarios through the incorporation of authentic educational events, achieving superior generalizability and ecological validity.

### 2.2 Multi-Agent Simulation of Human Interaction

In recent years, agentic systems based on large language models have become a research focus [15, 16]. Building on this trend, multi-agent simulation of human interaction has demonstrated significant potential in tasks such as personality analysis and social behavior research [17–19]. Early multi-agent systems primarily focused on two problem categories: goal-aligned collaborative tasks and game-theoretic competitive scenarios [20–22]. Moreover, a portion of the work is directed toward improving the agents' ability to engage in role-play [23–25]. Recent advancements have substantially expanded agent populations to investigate social dynamics, exemplified by Mou et al.'s work employing agent swarms to model opinion propagation in social networks [26] and Stanford's "Virtual Town" simulating complex human behavioral patterns through agent socialization [27]. However, practical applications leveraging multi-agent interaction for real-world problem-solving remain limited. Our proposed school-agent framework is specifically designed for synthesizing diverse, high-quality data to address this gap and meet operational requirements.

## 2.3 Knowledge Distillation

Knowledge distillation enables the transfer of knowledge from large teacher models to compact student models while preserving performance and reducing computational complexity. The seminal work by Hinton et al. first introduced this concept, demonstrating how soft labels could effectively compress models by distilling integrated knowledge from larger architectures [28]. Subsequent advancements by Jing et al. incorporated conditional generative adversarial networks to refine student logits outputs through adversarial training, achieving closer alignment with teacher outputs [29]. Zhang et al. expanded the paradigm through mutual learning among multiple student models, proving effective for collaborative training scenarios [30]. Novel directions emerged through Tung/Park et al.'s focus on relational similarity between teacher-student networks [31, 32], while Xu et al.'s alternating sampling method significantly narrowed performance gaps in complex mathematical reasoning tasks [33]. The recent "Branch-Merge Distillation" by DeepSeek successfully transferred DeepSeek-R1's capabilities to Qwen models across STEM benchmarks [34]. Our approach diverges by employing agent-mediated interaction to inject novel cognitive patterns into raw data, fundamentally overcoming knowledge distillation's inherent limitations in generalization capacity and teacher-dependent behavioral constraints through enhanced data format learnability.

# 3 Methodology

## 3.1 Overview

This section introduces our multi-agent classroom simulator, MACLASS. As illustrated in the Figure 1, MACLASS enables LLMs to realistically play the roles of teachers and students through carefully designed prompts and uses a set of original question–answer data as input to generate simulated teaching interaction scenarios. MACLASS integrates diverse real-world educational settings and adheres to the design principle of embedding authentic and effective pedagogical methods into the multi-agent interaction process. Our approach primarily addresses the following two challenges: (1) How can effective educational principles be incorporated into agent-based teaching processes? (2) How can we ensure coherent and natural interactions among agents across different scenarios?

To address the first challenge, we integrate established pedagogical principles into classroom interactions through three key approaches: teachers guide students to correct mistakes and solve problems independently, facilitating experiential learning from errors [35]; teachers facilitate group debates to enhance critical thinking and analytical skills [36]; and teachers encourage analogical reasoning, enabling students to develop inductive learning abilities by solving structurally similar problems [37]. The above methods are each developed into distinct instructional scenarios, featuring multi-turn interactions among multiple agents. The utterances from these agents are then concatenated and organized into ShareGPT-format data, effectively integrating foundational educational principles with the reasoning capabilities of intelligent agents within the dataset.

To address the second issue, we ensure that the interactions between multiple agent roles are coherently and logically controlled. To this end, we design precise interaction management rules that govern the speaking order of agents across different teaching scenarios, and assign scenario-specific prompts to each agent at different turns. Specifically, we model the dialogue process as: $\mathcal{D} = [\mathcal{D}_1, \mathcal{D}_2, \ldots, \mathcal{D}_i, \ldots, \mathcal{D}_n, \rho]$, where $D_i$ denote the utterance content of an agent at the $i$-th turn, and let $\rho$ represent the prompt used by the agent at the current teaching step. Under this framework, the predefined speaking order and prompt assignment strategy ensure the coherence of the dialogue flow, effectively achieving the intended pedagogical goals.

By adopting this method, we implement a fluent and well-controlled multi-agent classroom interaction module that successfully simulates authentic and effective teaching processes. The detailed configuration is provided in the Appendix.

## 3.2 Agent role construction

Studies have shown that interactive teaching can significantly enhance students' learning quality. However, its effective implementation relies on efficient dialogue exchanges between teachers and students, which remains a challenging task [38]. Simulating high-quality classroom interactions using multiple agents often encounters problems such as agents competing for roles, drifting off-topic,

or redundantly repeating previous responses. To address these issues, we introduce two strictly constrained types of agents: Teacher agents and Student agents. In our framework, we assign large language models (LLMs) with distinct prompts that enable them to assume different roles in a multi-agent setting or to perform different instructional tasks within the same role definition. This approach facilitates both functional cooperation and procedural control. Formally, this can be expressed as $\mathcal{A} = (\mathcal{L}, P_{R_i}), \quad R_i \in R = [R_1, R_2, \ldots, R_i, \ldots, R_n]$.

In our framework, each agent $\mathcal{A}$ powered by an LLM $\mathcal{L}$ is assigned role-specific prompts $P_{R_i}$ corresponding to distinct task phases $R_i$. For instance, a student agent utilizes different prompts when initially making an error versus when correcting it. This structured prompt design ensures agents operate within their designated roles and phases, minimizing role confusion and task redundancy, thereby enhancing the quality and stability of the collaboratively generated educational dialogues.

**Teacher Agents**  In a classroom environment, the teacher not only serves as the primary source of knowledge transmission and student guidance, but also plays a pivotal role in shaping the overall learning experience, managing instructional dynamics, and fostering critical thinking. As a core component of the multi-agent classroom framework, the teacher must simultaneously fulfill multiple instructional functions, including delivering content, assessing student understanding, providing timely feedback, and adapting pedagogical strategies to accommodate diverse learning needs.

Upon receiving an original question and its standard answer, the teacher agent conveys the problem details to the student, offering brief explanations to facilitate understanding. When provided with a student's solution, along with the corresponding question and standard answer, the teacher agent identifies any errors in the student's response and supplements the instruction with correct problem-solving strategies, guiding the student to independently rectify previous mistakes. This structured approach ensures the quality and stability of the multi-agent educational dialogues.

**Student Agents**  As the recipients of instructional content and the primary agents in the problem-solving process, students play a pivotal role in integrating the three pedagogical methods introduced in 3.1 into realistic educational scenarios.Student agents are expected not only to respond to teacher instructions but also to revise their previous answers based on prior attempts and peer debates, articulating their own perspectives accordingly.

## 3.3   Class specific settings

Our work creates a classroom environment that diverges from traditional instructional paradigms. Multi-agent systems that rely solely on predefined operations are insufficient to effectively simulate concrete pedagogical strategies. To authentically integrate the pedagogical methods outlined in 3.1 into the synthesized multi-agent classroom dialogue data, we have designed and developed a Classroom Interaction Manager comprising three modules: "Error Making and Correction", "Debate", and "Analogical Problem Retrieval and Solving". These modules respectively support the control of three distinct scenarios, as illustrated in Figure 1. The multi-turn question-answering augmented data generated from these three scenarios are concatenated in the ShareGPT format to construct a high-quality dataset named BOOST-QA.

**Error Correction Module**  This module is designed to simulate a classroom scenario where a student agent initially provides an incorrect or incomplete solution to a given problem. Subsequently, a teacher agent analyzes the student's response in conjunction with the standard answer to identify errors and offer correct reasoning. Finally, the student agent, leveraging the prior interaction and the standard answer, independently formulates a corrected solution. Specifically, we employ the Qwen2.5-0.5B-Instruct model with a temperature setting of 0.8 for the initial student response, promoting the generation of diverse and imperfect answers. For the subsequent correction phase, both teacher and student agents utilize the more capable Qwen2.5-14B-Instruct model with a temperature of 0.2, ensuring accurate error identification and high-quality reasoning.

In practice, training models with augmented data constructed by this module can inject structured noise into the gradient descent process. This simulates interactions across different cognitive levels, enabling the model to explore high-loss regions associated with student model errors and converge along smoother paths defined by corrected answers. Such an approach facilitates escaping local

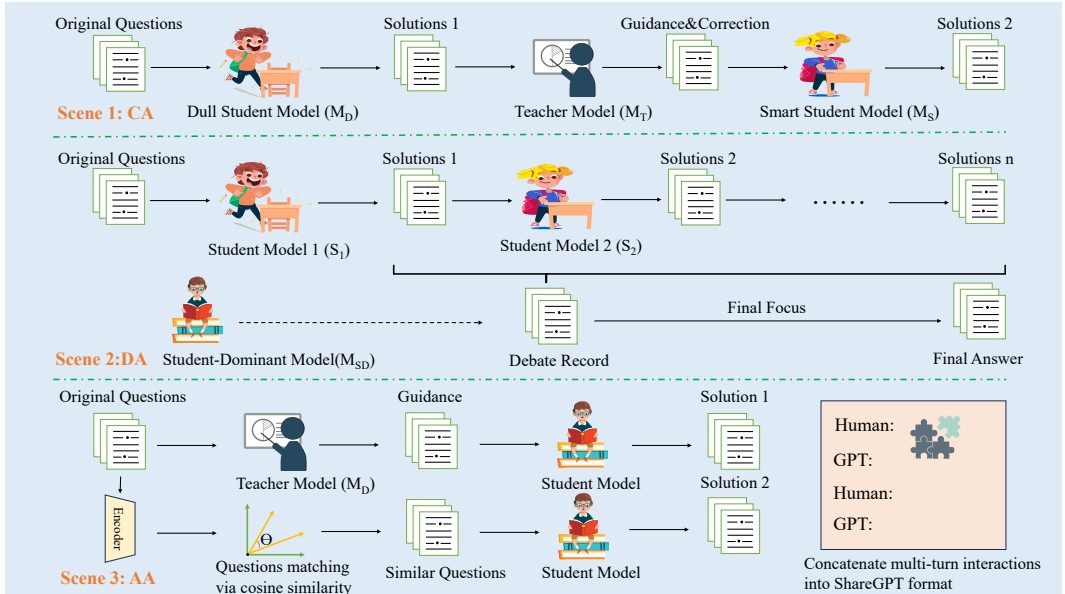

Figure 1: A multi-agent system-based data augmentation pipeline that simulates three different pedagogical contexts to enhance question-answering datasets. From top to bottom, the scenes are Correction Augmentation, Debatement Augmentation, and Analogical Augmentation.

minima and aids in identifying and avoiding common error patterns during inference. The gradient expression is as follows:

$$\nabla_\theta \mathcal{L}_{\mathrm{aug}}(\theta) \approx \nabla_\theta \mathcal{L}(\theta) + \gamma \cdot \mathbb{E}_{(x,\ \delta y) \sim \Delta \mathcal{D}} \big[ \nabla_\theta\ \ell\big( f_\theta(x),\ \delta y \big) \big] \tag{1}$$

In this context, $\mathcal{L}(\theta)$ denotes the conventional supervised learning loss function. The parameter $\gamma$ represents the perturbation intensity introduced by multi-agent interactions, influenced by factors such as inference temperature. The term $\mathbb{E}_{(x,\delta y) \sim \Delta \mathcal{D}}$ signifies the expectation over the augmented data distribution $\Delta \mathcal{D}$. Here, $\ell$ is the per-sample loss function, $f_\theta(x)$ corresponds to the corrected solution generated by the 14B teacher model for the original problem $x$, and $\delta y$ denotes the perturbed answer label produced by the 0.5B student model.

**Debate Interaction Module**   This module is designed to construct multiple student agents, each analyzing the question and the responses of other students to express their own perspectives. This approach aims to capture diverse problem-solving strategies and enhance data diversity. We employ identical prompts to build three student agents: $S_1$, $S_2$, and $S_3$. $S_3$ utilizes the Qwen2.5-14B-Instruct model with a temperature setting of 0.2, serving as the summarizer and decision-maker in the debate. Conversely, $S_1$ and $S_2$ employ the Qwen2.5-7B-Instruct model with a temperature of 0.6, acting as regular participants in the discussion. During the data augmentation process, $S_1$ and $S_2$ take turns speaking for one or two rounds, after which $S_3$ provides a summary. The output of the agent in the i-th round, denoted as $A_i$, can be represented by $A_i = f(H, \theta_i)$, where $H$ signifies the current classroom dialogue history, and $\theta_i$ denotes the output temperature corresponding to the agent.

Analogous to the error correction module, this component employs discussions among multiple agents to simulate the cognitive diversity of students. By fine-tuning the model with debate-enhanced data, it facilitates the smoothing of the loss landscape, thereby guiding the optimization trajectory away from sharp local extrema and enabling the learning of diverse, high-quality problem-solving strategies. The gradient representation of this module is as follows:

$$\nabla_\theta \mathcal{L}_{\text{aug}} = \mathbb{E}_{(x,y)\sim\mathcal{D}} \left[ \underbrace{\nabla_\theta \ell(f_\theta(x), y)}_{\text{original gradient}} + \lambda \underbrace{\sum_{k=1}^{K} \nabla_\theta \ell(f_\theta(x), y_k^*)}_{\text{debate gradient}} \right]. \tag{2}$$

Here, $\mathbb{E}_{(x,y)\sim\mathcal{D}}$ denotes the expectation over the augmented data distribution $\mathcal{D}$. The term $\nabla_\theta \ell(f_\theta(x), y)$ represents the original gradient. The parameter $\lambda$ is the debate intensity coefficient, influenced by factors such as model inference temperature and the number of debate rounds. $K$ denotes the number of debate rounds, $y_k^*$ is the answer provided by the agent in the current round, and $\ell(f_\theta(x), y_k^*)$ represents the single-sample loss.

**Similar Question Retrieval Module**    This module facilitates basic interactions between teacher and student agents. After the student completes the first-round response, the system retrieves similar questions based on the initial prompt. By constructing new prompts for the student to solve related problems and subsequently performing format concatenation, the system builds analogy-based data. This approach encourages the application of existing knowledge to analogous scenarios, enhancing knowledge transfer capability.

Specifically, a small subset of questions is randomly selected from the original dataset as the first-round questions $Q_{1st}$, while the remaining questions form the retrieval pool. Using the all-MiniLM-L6-v2 model, embeddings are computed, and cosine similarity is employed to identify questions in the pool that closely resemble $Q_{1st}$. From these, one question is randomly chosen as the analogy reasoning question $Q_{2nd}$ for the student's second-round response, as illustrated in the accompanying formula. Both student and teacher agents utilize the Qwen2.5-14B-Instruct model with a temperature setting of 0.2. $Q$ denotes the complete question dataset, and $Q_{1st}^c$ represents the complement set of first-round questions, the analogy retrieval process is as follows:

$$Q_{2st} = \text{Random}\left(\text{Top-k}\left(\cos\_\text{sim}(Q_{1st}, Q_{1st}^c)\right)\right), \quad Q_{1st} \cup Q_{1st}^c = Q. \tag{3}$$

Specifically, we employ the ShareGPT format to concatenate two rounds of question-answer dialogues from multi-agent students addressing similar problems into a single mixed training sample. This approach facilitates joint modeling of locally similar samples within the semantic space during model training, effectively serving as an implicit interpolation-based augmentation. Moreover, this mechanism encourages the model to focus on subtle differences between similar samples, thereby learning the core semantics of the task and enhancing robustness. The loss function for training the model with this augmented data is defined as follows:

$$\mathcal{L}_{\text{mix}} = -\log p(y_1, y_2 \mid x_1, x_2). \tag{4}$$

Here, $x_1$ and $x_2$ denote the original problem texts, while $y_1$ and $y_2$ represent the corresponding original solution texts. The formula thereby signifies the process of learning the joint probability distribution between similarly structured and closely related sample pairs.

## 4  Experiments

### 4.1  Experimental Setup

**Training datasets.**    We used three instruction-tuning datasets: (1) Orca-Math-Word-200K, a high-quality set of elementary math QA pairs generated via multi-agent collaboration [39]; (2) ProcQA, mixed-modality programming QA pairs from StackOverflow [40]; and (3) OpenHermes 2.5, a general-purpose dataset covering commonsense QA and reasoning. We sampled 10,000 instances each from Orca-Math-Word-200K and ProcQA, and 9,000 from OpenHermes 2.5, forming the original dataset (ori-data). Applying the MASTER augmentation method to ori-data produced an equal-sized enhanced dataset (19,000 samples), termed BOOST-QA. Correctness verification with a locally deployed Qwen2.5-32B-Instruct model showed only **4.1%** of augmented samples contained procedural reasoning errors.

**Evaluation datasets.** We evaluated our method on HumanEval [41], MBPP [42], MATH [43], MMLU-PRO-MATH [44], MMLU [45], ARC [46] and SCI-Q[47]. These datasets encompass various domains and task types, including human-written coding challenges, mathematical problem-solving, multi-choice questions, and scientific reasoning, thereby providing a comprehensive assessment of our method's capabilities. During evaluation, we assessed the zero-shot capabilities of the MASTER-model series across these datasets. The inference temperature was set to 0 for HumanEval and MBPP, and to 0.2 for all other datasets.

**Models for data augmentation and training.** We employed the Qwen2.5-Instruct series of models [48] as the foundational models to enhance the original data through multi-agent classroom interactions; specific configurations are detailed in 3.3 . To evaluate the effectiveness of MASTER, we utilized three base models: LLaMA-3-7B-base [49], Qwen2.5-7B-base [48], and Mistral-7B-base [50]. By fine-tuning these models on BOOST-QA, ori-data, and other high-quality datasets constructed using alternative methods, and subsequently comparing their performances, we validated the superiority of the MASTER method in enhancing data quality.

## 4.2 Baselines

We selected four baseline methods for comparison with MASTER. First, we employed traditional text augmentation techniques by injecting character-level noise into the original text. Inspired by EDA, RandomAug and SpellingAug [51] , were selected and are both open-sourced in the GitHub project nlpaug.

The third baseline is TAGCOS [52], which computes gradient representations for each sample in the original dataset, clusters similar data points, and then applies a greedy algorithm within each cluster to effectively select high-quality data subsets for instruction fine-tuning. This approach emphasizes efficiency and relevance in data selection, aiming to reduce redundancy while preserving diversity in the fine-tuning corpus.

The final baseline is CoT Collection, proposed by Seungone Kim et al., which aims to enhance the reasoning capabilities of small and medium-sized language models in zero-shot and few-shot tasks through chain-of-thought (CoT) fine-tuning [53]. It aggregates a large and diverse set of CoT-annotated samples from multiple sources to provide explicit reasoning supervision, thereby helping models better learn intermediate reasoning steps.

To ensure a fair comparison, we used the TAGCOS method to select high-quality subsets of 5K, 5K, and 9K samples from the original training datasets Orca-Math-200K, ProcQA, and OpenHermes2.5, respectively, and combined them into a new high-quality training set. RandomAug, SpellingAug, CoT Collection, and our MASTER were then applied to randomly augment an equal amount of original data. Each method was used to fine-tune the pretrained model using LoRA for 2 epochs with a learning rate of 1e-4.

## 4.3 Main Results

We present the primary results of different models trained on BOOST-QA and Ori-Data across various benchmarks in Table 1 and Table 2, and compare them with multiple baseline methods in Table 3. We use accuracy as the evaluation metric for mathematics and general tasks. For objective questions, answers are extracted reliably using regular expressions, while subjective questions are evaluated for correctness by the Qwen2.5-14B-Instruct model based on the corresponding reference answers. For programming tasks such as HumanEval and MBPP, we adopt Pass@1 as the primary evaluation metric. Our findings are as follows:

**BOOST-QA has demonstrated performance improvements across various models.** In the experimental results presented in Table 1 and Table 2, all baseline models fine-tuned with the augmented dataset BOOST-QA generally outperformed those fine-tuned with the original, unaugmented dataset Ori-Data across multiple benchmark tests. This indicates that our data augmentation method effectively enhances the learnability of the original data, thereby improving the model's generalization ability for the tasks.

**The BOOST-QA dataset enabled models to achieve better performance compared to other baselines.** In the experimental results presented in Table 3, the LLaMA3-8B-base model fine-tuned with BOOST-QA dataset consistently outperforms a wide range of baseline methods across multiple benchmarks in mathematics, programming, and general tasks. This highlights the significant advantages of our MASTER data augmentation method.

Table 1: Performance comparison of models fine-tuned with Ori-Data and BOOST-QA (Part 1).

| Model | MATH | | MMLU-PRO-MATH | | MBPP | | HumanEval | |
|---|---|---|---|---|---|---|---|---|
| | Ori | BOOST-QA | Ori | BOOST-QA | Ori | BOOST-QA | Ori | BOOST-QA |
| LLaMA3-8B | 21.58 | **23.90** | 13.55 | **27.39** | 65.30 | **67.20** | 39.02 | **50.61** |
| Qwen2.5-7B | **71.00** | 70.54 | 24.35 | **44.41** | 78.00 | **79.10** | 22.56 | **42.07** |
| Mistral-7B | 15.74 | **17.58** | 7.18 | **13.92** | **56.30** | 55.00 | 17.68 | **28.05** |

Table 2: Performance comparison of models fine-tuned with Ori-Data and BOOST-QA (Part 2).

| Model | MMLU | | ARC | | SCI-Q | | AVERAGE | |
|---|---|---|---|---|---|---|---|---|
| | Ori | BOOST-QA | Ori | BOOST-QA | Ori | BOOST-QA | Ori | BOOST-QA |
| LLaMA3-8B | 48.13 | **48.13** | 57.76 | **61.52** | 76.50 | **80.10** | 45.98 | **51.26** |
| Qwen2.5-7B | 24.05 | **50.12** | 20.90 | **68.52** | 20.00 | **69.70** | 37.27 | **60.64** |
| Mistral-7B | 27.59 | **35.89** | 32.25 | **47.70** | **70.50** | 49.10 | 32.46 | **35.32** |

Table 3: Performance comparison of models fine-tuned with BOOST-QA and other baselines.

| Method | Ori | RandomAug | SpellingAug | TAGCOS | CoT-fine | BOOST-QA |
|---|---|---|---|---|---|---|
| MATH | 21.58 | 21.16 | 20.86 | **26.50** | 21.02 | 23.90 |
| MMLU-PRO-MATH | 13.55 | 14.58 | 13.40 | 12.95 | 14.36 | **27.39** |
| MBPP | 65.30 | 61.90 | 63.50 | 61.40 | 61.40 | **67.20** |
| HumanEval | 39.02 | 31.10 | 37.20 | 42.68 | 33.54 | **50.61** |
| MMLU | 48.13 | 38.35 | 24.21 | 46.91 | 41.08 | **48.13** |
| ARC | 57.76 | 41.98 | 22.61 | 61.09 | 47.87 | **61.52** |
| SCI-Q | 76.50 | 62.30 | 22.70 | **84.00** | 68.50 | 80.10 |
| Average | 45.98 | 38.77 | 29.21 | 47.93 | 41.11 | **51.26** |

**The BOOST-QA dataset significantly enhances the model's capability in solving complex multiple-choice questions.** Notably, We found that our MASTER series models achieved remarkable improvements on multiple-choice question tasks. Upon observing this phenomenon, we incorporated additional multiple-choice benchmark tests, with the results illustrated in Figure 2. Across the current eight test datasets, the MASTER method demonstrated consistent improvements exceeding 5% compared to training with the original data, peaking at a 31.46% enhancement. Through analysis of the models' inference outputs, we found that the MASTER-series models generated significantly longer reasoning chains than those trained on the original data, where outputs tended to be shorter or limited to direct option selection. This indicates that our BOOST-QA dataset effectively internalizes the models' reasoning capabilities, enabling better generalization when addressing complex problems.

### 4.4 Ablation Study

In this section, we meticulously design ablation experiments by fine-tuning the LLaMA3-8B-base model using training data constructed from various combinations of educational scenarios. This approach is intended to assess the impact of different teaching scenarios on model performance.

**Effects of two scenarios** In a subsequent series of ablation experiments, as detailed in Table 4, we explored the impact of combining different pairs of educational scenarios on model performance. Specifically, ME&DB (Make Error and Debate), ME&EP (Make Error and Expand), and DB&EP (Debate and Expand) denote models fine-tuned with augmented data that integrates these scenario pairs. To construct these dual-scenario datasets, we meticulously removed data lacking the corresponding scenarios from BOOST-QA and supplemented it with carefully selected samples from

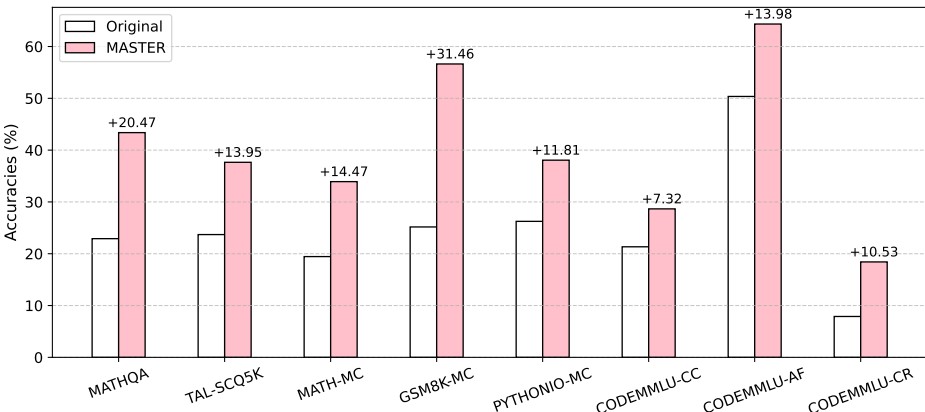

Figure 2: Results from the complex multiple-choice question test show a maximum improvement of 31.46% and an average improvement of 15.50%. CODEMMLU-CC, CODEMMLU-AF, and CODEMMLU-CR are abbreviations for CODEMMLU-CODE-COMPLETION, CODEMMLU-API-FRAMEWORKS, and CODEMMLU-CODE-REPAIR, respectively.

Table 4: Performance comparison of models fine-tuned with partial and full multi-agent simulated teaching scenarios across multiple benchmarks. ME, DB, and EP are the abbreviations for Make Error, Debate, and Expand, respectively.

| Method | Ori | ME&DB | ME&EP | DB&EP | ME | DB | EP | Full |
|--------|-----|-------|-------|-------|-----|-----|-----|------|
| MATH | 21.58 | 19.88 | 19.90 | 23.48 | 17.96 | 21.90 | 22.40 | **23.90** |
| MMLU-PRO-MATH | 13.55 | 16.14 | 17.25 | 17.62 | 13.40 | 11.32 | 10.58 | **27.39** |
| MBPP | 65.30 | 56.10 | 52.90 | 66.10 | 52.90 | 63.50 | 61.60 | **67.20** |
| HumanEval | 39.02 | 31.70 | 36.59 | 45.73 | 23.78 | 47.56 | 44.51 | **50.61** |
| MMLU | 48.13 | 40.08 | 35.36 | 32.64 | 40.54 | 19.86 | 31.40 | **48.13** |
| ARC | 57.76 | 49.57 | 41.13 | 32.34 | 47.35 | 50.26 | 38.14 | **61.52** |
| SCI-Q | 76.50 | 56.30 | 63.60 | 55.20 | 67.90 | 54.40 | 50.20 | **80.10** |
| Average | 45.98 | 38.54 | 38.10 | 39.02 | 37.69 | 38.40 | 36.98 | **51.26** |

Ori-Data. The experimental results indicate that augmenting with only one or two scenarios fails to significantly enhance model performance. In contrast, the MASTER model, which is trained with data augmented from all three educational scenarios, consistently outperforms the model trained on the original data across all test sets. This finding underscores the complementary and indispensable roles of each educational scenario in the data augmentation process, highlighting that a combination of multiple scenarios is essential for optimal model performance.

**Effects of one scenarios** In Table 4, we systematically evaluated the performance of models trained on datasets augmented using only a single educational scenario—specifically, ME (Make Error), DB (Debate), or EP (Expand). The results consistently showed that these models underperformed compared to the model trained solely on the original dataset across all test sets. This outcome highlights the limitations of augmenting data with a single scenario, as it fails to provide the comprehensive learning experiences necessary for robust model performance. Furthermore, the models trained with only one scenario exhibited a lack of robustness when faced with diverse test conditions, indicating that a singular approach is insufficient for comprehensive learning. Our findings suggest that integrating multiple educational scenarios is crucial for enhancing the adaptability and generalizability of the models, as each scenario contributes unique learning signals that collectively improve model performance. Therefore, we conclude that a combination of varied educational scenarios is essential for effectively improving model performance, as it provides a more holistic and diverse learning environment.

# 5 Conclusion

This study systematically investigates the impact of constructing multi-agent instructional scenarios on question-answering (QA) data augmentation. To obtain high-quality instruction fine-tuning data, we simulate three distinct educational scenarios using multiple agents, introducing varying levels of cognitive interaction into the original data. This approach aims to enhance the convergence efficiency of the base model on the augmented data. Furthermore, during the inference phase, we implement an error-correction interaction pattern that mirrors the structure of the training data, ensuring consistency between training and reasoning processes. Our experimental results validate the effectiveness of this comprehensive framework in improving model performance.

## Acknowledgments and Disclosure of Funding

We thank the reviewers for their constructive comments. This work was supported in part by the National Natural Science Foundation of China under Grant 62350710797, Grant 62276077, Grant 62406091, and Grant U23B2055, in part by CCF-Baidu Open Fund (CCF-Baidu 202404), in part by Guangdong Basic and Applied Basic Research Foundation under Grant 2024A1515011205, and in part by the Shenzhen Science and Technology Program under Grant KQTD20240729102154066 and Grant ZDSYS20230626091203008.

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

# Appendices

## A Author Contributions

Liang Yue designed and implemented a detailed multi-agent interaction process, conducted extensive experiments, refined the training data augmentation, supervised fine-tuning (SFT), and evaluation procedures, and successfully reproduced certain baseline methods, thereby making a significant contribution to the manuscript. Yihong Tang proposed effective suggestions, developed a method leveraging large language models (LLMs) of varying sizes as interactive agents to enrich the data space, guided the design of fine-tuning and analytical experiments, and contributed substantially to the manuscript. Kehai Chen, Jie Liu and Min Zhang jointly led the project, oversaw team resource allocation and collaboration, and offered valuable advice on experiments and writing, thus making important contributions.

## B Limitations and Future Work

**Generalization** We have observed that augmenting training data using the MASTER method significantly enhances large language models' (LLMs) performance on complex reasoning tasks. This improvement is likely due to the method's focus on generating training data with coherent thought processes, thereby strengthening the models' logical reasoning capabilities. However, it's important to note that such enhancements are relatively limited in tasks like commonsense question answering and reading comprehension. These tasks often rely more heavily on extensive background knowledge and contextual understanding rather than deep logical reasoning. Consequently, the application of multi-agent data augmentation mechanisms in commonsense and reading comprehension tasks warrants further exploration and optimization to achieve more substantial improvements.

**Applicability** Due to constraints in computational resources and time, we constructed a relatively small training dataset and validated the effectiveness of our method using pre-trained models with fewer than 8 billion parameters. Our findings indicate that BOOST-QA data significantly enhances the performance of these smaller models compared to the original training data. However, considering that large-parameter models, which have undergone extensive training on diverse datasets, and instruction-tuned models trained on carefully curated instruction sets, may have already internalized strong reasoning capabilities, our method might yield less pronounced improvements for them. Therefore, future work could explore whether multi-agent teaching and data augmentation strategies can effectively enhance performance in large-parameter instruction-tuned models.

**Is it necessary to train a teacher-student model?** In our multi-agent interaction framework, we employed the Qwen2.5 series models, which have demonstrated strong performance in mathematical and programming tasks, to serve as both teacher and student agents. However, these models were not specifically trained for error correction and debate-style interactions. Therefore, future work could involve constructing a small set of high-quality dialogue data tailored to these scenarios, enabling the training of specialized teacher and student models. This approach may yield more effective outcomes in such interactive settings.

**How to maintain data security and positivity?** In the error correction scenarios, we utilized a 0.5B parameter model as the student agent to intentionally generate errors, with explicit prompts indicating its limited capabilities. However, the outputs from this "underperforming" model may negatively influence the subsequent error correction process between the teacher and student agents. Additionally, there is a risk that the large language model (LLM) might replicate irrelevant or inappropriate outputs from the student agent, deviating from the intended context. To mitigate these issues, future research could incorporate a classroom management agent and implement data screening mechanisms to ensure that the generated training data is safe, accurate, and aligns with positive educational objectives.

> Human: You want to multiply 20 by some number. I mistakenly multiplied 10 by the number, and the result was 40. What is the correct calculation result?
>
> GPT(Dull Student): To find the correct calculation result, let's break down the problem:\n\n1. **Identify the mistake**: I mistakenly multiplied 10 by the unknown number, and the result was 40. This means that the unknown number should be:\n  - Since 10 * 40 = 400, and 20 * 40 = 800, the correct unknown number should be the inverse of this, i.e., 800 / 20 = 40.\n\n2. **Correct calculation**: Given the correct multiplication:\n  - 20 * 40 = 800\n  - 10 * (40 / 20) = 10 * 2 = 20\n\n3. **Final step**: Multiply the correct result by 20:\n  - 800 * 20 = 16000\n\nSo, the correct calculation result is 16000.
>
> Human(Teacher): Your solution process is incorrect. Let's focus on the first step where you identified the mistake. You need to find the unknown number that, when multiplied by 10, gives 40. Once you have the correct unknown number, you can then multiply it by 20 to get the correct result. Reconsider how to find the unknown number from the given information.
>
> GPT(Smart Student): I see where I went wrong. Let's correct it step by step:\n\n1. **Identify the unknown number**: Since multiplying 10 by the unknown number gives 40, the unknown number is:\n  - 40 / 10 = 4\n\n2. **Correct calculation**: Now, multiply the correct unknown number by 20:\n  - 20 * 4 = 80\n\nSo, the correct calculation result is 80.

Figure 3: Example of Multi-Agent Error Correction Scenario Augmented Data in ShareGPT Format

> Human: John wants to finish a show in 5 days. There are a certain number of episodes and they are each 30 minutes long. He has to watch 2 hours a day. How many episodes are in the show?
>
> GPT(Student A): To determine the total number of episodes in the show, we first need to calculate how many minutes John will watch in total over the 5 days. Since he plans to watch 2 hours each day, and there are 60 minutes in an hour, we can calculate the total minutes as follows:\n\n2 hours/day * 60 minutes/hour * 5 days = 600 minutes\n\nGiven that each episode is 30 minutes long, we can now find out the total number of episodes by dividing the total minutes by the length of each episode:\n\n600 minutes / 30 minutes/episode = 20 episodes\n\nTherefore, the show has 20 episodes in total.
>
> Human(Student B): The calculation provided by StudentA seems correct. John plans to watch 2 hours each day, which is 120 minutes per day. Over 5 days, he will watch a total of 600 minutes. Since each episode is 30 minutes long, dividing 600 minutes by 30 minutes per episode indeed gives us 20 episodes. However, I would like to recheck the calculation to ensure there are no mistakes. Let's break it down again:\n\nFirst, calculate the total minutes John will watch over 5 days:\n2 hours/day * 60 minutes/hour * 5 days = 600 minutes\n\nNext, divide the total minutes by the length of each episode:\n600 minutes / 30 minutes/episode = 20 episodes\n\nThe result is the same as StudentA's, so the show has 20 episodes in total.
>
> GPT(Student C): The calculation provided by StudentA and StudentB is indeed correct. John plans to watch 2 hours each day, which amounts to 120 minutes per day. Over 5 days, he will watch a total of 600 minutes (2 hours/day * 60 minutes/hour * 5 days). Since each episode is 30 minutes long, dividing the total minutes by the length of each episode gives us 20 episodes (600 minutes / 30 minutes/episode). Therefore, the show has 20 episodes in total.

Figure 4: Example of Multi-Agent Debate Scenario Augmented Data in ShareGPT Format

## C   Analysis and Visualization of Augmented Data

The enhanced example data formats for the error correction, debate, and analogical reasoning scenarios are illustrated in Figure 3, Figure 4 and Figure 5 respectively. Each scenario's agent outputs are concatenated to create training data in the ShareGPT format.

## D   Detailed Experimental Design

### D.1   Train Data

To evaluate the effectiveness of the MASTER data augmentation method, we constructed six training datasets, each comprising 19,000 instances formatted in the ShareGPT style. These datasets include:

Human(Teacher): Tommy goes for a run around his neighborhood and decides to calculate how many wheels he saw. All the vehicles in his neighborhood have 4 wheels. He saw 12 trucks and some cars, and he saw 100 wheels. How many cars did he see?The question asks you to determine the number of cars Tommy saw based on the total number of wheels he counted and the number of trucks he observed. Given that each vehicle has 4 wheels, and Tommy saw a total of 100 wheels from 12 trucks and an unknown number of cars, you need to set up an equation to represent this scenario. Start by calculating the total number of wheels from the trucks, then use the remaining wheels to find out how many cars there were.

GPT(Student): To determine the number of cars Tommy saw, we start by noting that each vehicle has 4 wheels. Tommy saw 12 trucks, so the total number of wheels from the trucks is $12 \times 4 = 48$ wheels. Since Tommy saw a total of 100 wheels, the remaining wheels must have come from the cars. The number of wheels from the cars is $100 - 48 = 52$. Since each car also has 4 wheels, the number of cars Tommy saw is $52 \div 4 = 13$. Therefore, Tommy saw 13 cars.

Human(Teacher): Good job,please take a look at this similar question.Tommy goes for a run around his neighborhood and decides to calculate how many wheels he saw. All the trucks in his neighborhood have 4 wheels, all the cars have 4 wheels, bicycles have 2 wheels, and buses have 6 wheels. If he saw 12 trucks, 13 cars, 8 bicycles, and 3 buses, how many wheels did he see?

GPT(Student): To calculate the total number of wheels Tommy saw, we need to multiply the number of each type of vehicle by the number of wheels they have and then add all the results together. For trucks, there are 12 trucks each with 4 wheels, so 12 trucks * 4 wheels/truck = 48 wheels. For cars, there are 13 cars each with 4 wheels, so 13 cars * 4 wheels/car = 52 wheels. For bicycles, there are 8 bicycles each with 2 wheels, so 8 bicycles * 2 wheels/bicycle = 16 wheels. For buses, there are 3 buses each with 6 wheels, so 3 buses * 6 wheels/bus = 18 wheels. Adding all these wheels together, we get 48 wheels (trucks) + 52 wheels (cars) + 16 wheels (bicycles) + 18 wheels (buses) = 134 wheels. Therefore, Tommy saw a total of 134 wheels during his run.

Figure 5: Example of Multi-Agent Analogical Reasoning Scenario Augmented Data in ShareGPT Format

(1) the original instruction dataset; (2) an instruction dataset augmented using the MASTER method; (3) datasets enhanced through two Easy Data Augmentation (EDA) techniques, namely RandomAug and SpellingAug; (4) an instruction dataset augmented via Chain-of-Thought (CoT) prompting; and (5) a high-quality instruction dataset curated from multiple sources using the TAGCOS method. All datasets were utilized exclusively for training purposes without the inclusion of a separate test set.

## D.2 Train Setup

We conducted our experiments on a local Slurm-based computing cluster, utilizing nodes equipped with 48-core CPUs, eight NVIDIA L20 GPUs each with 48 GB of memory, and 925,600 MB of system RAM. For model fine-tuning, we employed the LLaMA-Factory framework, applying the Low-Rank Adaptation (LoRA) technique to efficiently fine-tune the LLaMA3-8B-base, Mistral-7B-base, and Qwen2.5-7B-base models. Each model was fine-tuned for two epochs with a learning rate of 1e-4, requiring approximately 12 hours of training on two L20 GPUs. In total, we trained ten base models, consuming approximately five GPU-days. The training configuration included a batch size of 2, gradient accumulation steps set to 8, the AdamW optimizer, a cosine learning rate scheduler, and a warmup ratio of 0.1.

## D.3 Math Evaluation

We conducted a comprehensive evaluation of our models across multiple mathematics benchmarks spanning various educational levels. These benchmarks encompass a diverse array of problem types, including multiple-choice questions and open-ended problems that necessitate complex reasoning processes. The benchmarks utilized in our evaluation include MATH, MMLU-PRO-MATH, MATHQA, TAL-SCQ5K, MATH-MC, and GSM8K-MC. For assessment, we employed accuracy as the primary metric. To evaluate open-ended questions, we utilized the Qwen2.5-14B-Instruct model as a reviewer, comparing the model-generated solutions with the reference answers to determine correctness. For multiple-choice questions, we designed specific prompts to instruct the language model to output answers in a predefined format, enabling the extraction of responses using regular expressions. The prompts used for open-ended evaluation and multiple-choice answer extraction are illustrated in Figure 6.

```
Prompt for objective questions:
user_prompt = f"""<Teacher> Question: {problem}
Options: {options_str}

Please select the answer from {', '.join(option_labels)}. Finally, provide your answer in the format [x], where x is
the index of the correct option."""

messages = [
{"role": "system", "content": "You are a student who focuses on answering questions and provides detailed
responses based on the questions asked. "},
{"role": "user", "content": user_prompt}
]
Prompt for subjective evaluation:
user_prompt = f"""Question: {problem}
Standard answer: {answer}
LLM answer: {llm_answer.get(student_field, ")}

Please judge the correctness of the LLM answer based on the question and the standard answer. If it is correct,
output a <1> at the end, and if it is wrong, output a <0> at the end."""
messages = [
{"role": "system", "content": "You are a strict math teacher and you need to judge the correctness of LLM's
answers based on the questions and standard answers. "},
{"role": "user", "content": user_prompt}
]
```

Figure 6: Examples of prompts used for evaluating objective and subjective benchmark tasks. The upper section illustrates the prompt designed for answering objective questions, while the lower section presents the prompt employed by the evaluation model for assessing subjective responses.

## D.4 Code Evalution

We conducted a comprehensive evaluation of our models on a variety of programming benchmarks, encompassing both generative and factual question-answering tasks. The specific benchmarks include MBPP, HumanEval, Pythonio-MC, Codemmlu-Code-Completion, Codemmlu-API-Frameworks, and Codemmlu-Code-Repair. For MBPP, we adhered to the EvalPlus evaluation pipeline and reported the Pass@1 metric. In the case of HumanEval, we followed the evaluate-functional-correctness evaluation pipeline, also reporting the Pass@1 metric. Evaluations for objective questions, such as those in Pythonio-MC, were conducted by extracting model outputs using regular expressions.

## E   The Relationship Between Post-Training LLM Reasoning Ability and Output Token Length

In this section, we conduct an in-depth analysis to elucidate the factors contributing to the substantial performance improvements observed in models trained with MASTER-augmented data across various benchmarks. Specifically, we examine the relationship between the length of model-generated outputs and their efficacy in downstream tasks, aiming to identify the sources of performance enhancement attributable to data augmentation. Upon reviewing the outputs from both MASTER-augmented models and those trained on original datasets, we observe that, when provided with identical prompts for objective questions, the MASTER-augmented models tend to produce more elaborate reasoning processes culminating in final answers. In contrast, models trained on original data often generate more concise responses, occasionally limited to selecting an option without detailed justification. Representative examples illustrating the distinct reasoning approaches of these two model types are presented in Figure 7.

Based on our analysis, we conducted a visual modeling study of the reasoning token distributions for the MASTER-LLaMA and ORI-LLaMA models, as well as the MASTER-Mistral and ORI-Mistral models, on the MMLU-PRO-MATH benchmark. As depicted in Figure 8, it is evident that, when provided with identical prompts, the MASTER series models engage in more extensive and detailed reasoning processes, resulting in longer outputs and higher accuracy rates compared to their counterparts trained on original data.

MASTER-Augmented Data Model Output Example:

Example1: Statement 1: Any set of two vectors in R^2 is linearly independent.\nThis statement is false because a set of two vectors in R^2 can be linearly dependent. For example, consider the vectors (1, 0) and (0, 1). These vectors are linearly dependent because (1, 0) = 1 * (1, 0) + 0 * (0, 1).\n\nStatement 2: If V = span(v1, ... , vk) and {v1, ... , vk} are linearly independent, then dim(V) = k.\nThis statement is true. If a set of vectors is linearly independent, then the vectors are linearly independent in the span of those vectors. Therefore, the dimension of the span is equal to the number of linearly independent vectors.\n\nThus, the correct answer is [C].

Example2: [C] Not enough information, True\nThe statements provided are not sufficient to determine whether every permutation is a cycle and every cycle is a permutation. Statement 1 asserts that every permutation is a cycle, which is a true statement in the context of permutations. However, Statement 2 claims that every cycle is a permutation, which is not universally true. For instance, a cycle can be a permutation, but not every cycle is a permutation. Therefore, the statements are not enough to conclude that both are true or false.

Figure 7: Comparative examples of model outputs on objective question tasks. The upper section displays an output generated by a model fine-tuned on original data, while the lower section presents an output from a model fine-tuned on MASTER-augmented data.

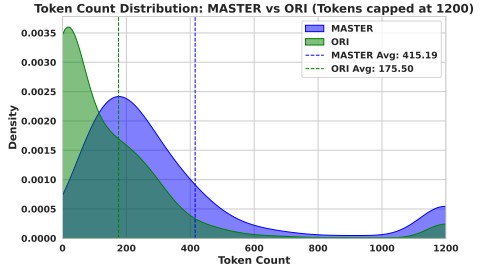

(a) Reasoning Token Length Distribution of the ORI-LLaMA-Model and MASTER-LLaMA-Model

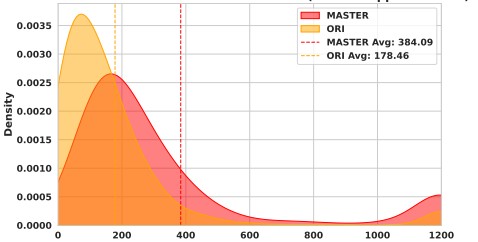

(b) Reasoning Token Length Distribution of the ORI-Mistral-Model and MASTER-Mistral-Model

Figure 8: Kernel Density Estimation (KDE) plots illustrating the token count distributions for model outputs on the MMLU-PRO-MATH benchmark. Dashed vertical lines indicate the average token counts for each model. Token counts are capped at 1,200 to mitigate the influence of outliers.

We further conducted a binned visualization analysis of sample counts and accuracy across different token length intervals, as illustrated in Figure 9, Figure 10, Figure 11, Figure 12. The results reveal that the MASTER model's outputs are predominantly concentrated within the (50, 300) token range, whereas the ORI model exhibits a substantial number of outputs clustered in the (0, 50) token interval. Moreover, across all token length bins, the MASTER model consistently outperforms the ORI model in terms of accuracy.

# F    Prompt Engineering for Multi-Agent Systems

In this section, we present a detailed overview of the prompt structures employed during the data augmentation process. Specifically, Table 5, Table 6, Table 7 illustrate the agent prompts designed for error correction, debate, and analogy scenarios, respectively.

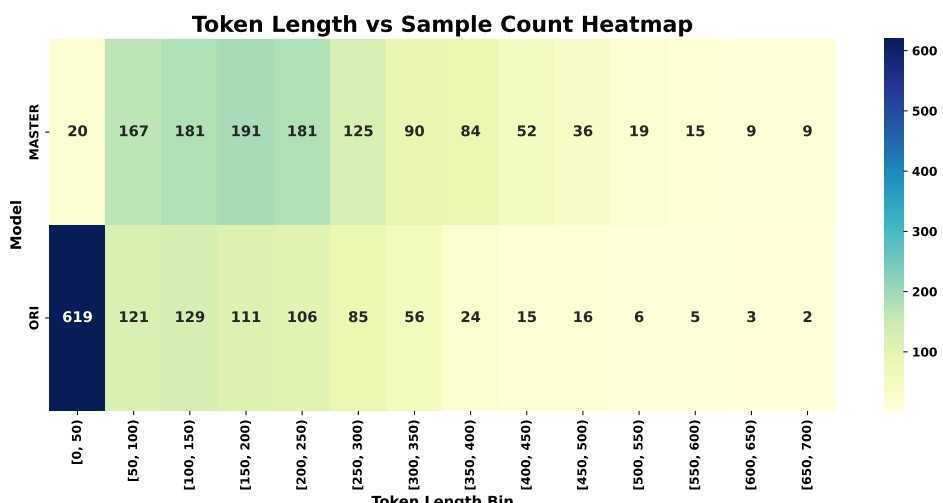

Figure 9: Boxplot of Inference Token Lengths for the MASTER-LLaMA Model on the MMLU-PRO-MATH Benchmark, illustrating the Distribution of Sample Outputs Across Different Token Lengths.

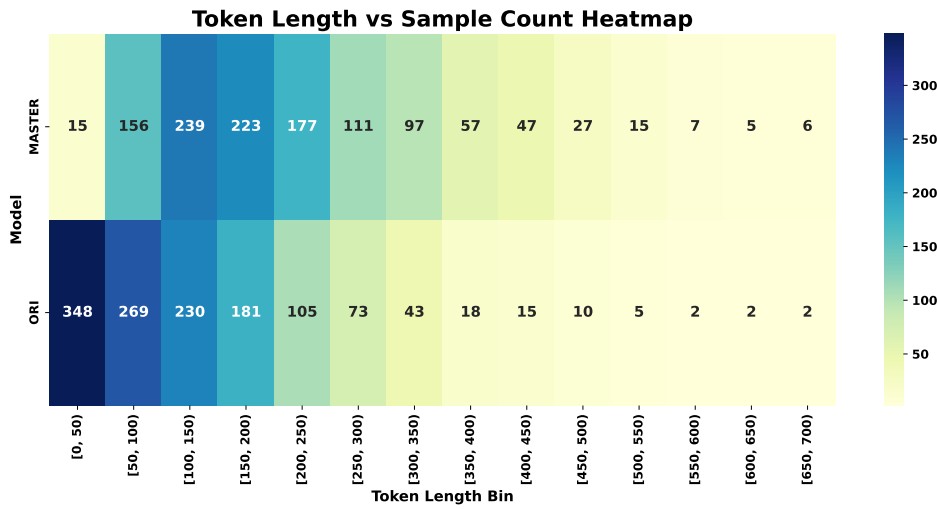

Figure 10: Boxplot of Inference Token Lengths for the MASTER-Mistral Model on the MMLU-PRO-MATH Benchmark, illustrating the Distribution of Sample Outputs Across Different Token Lengths.

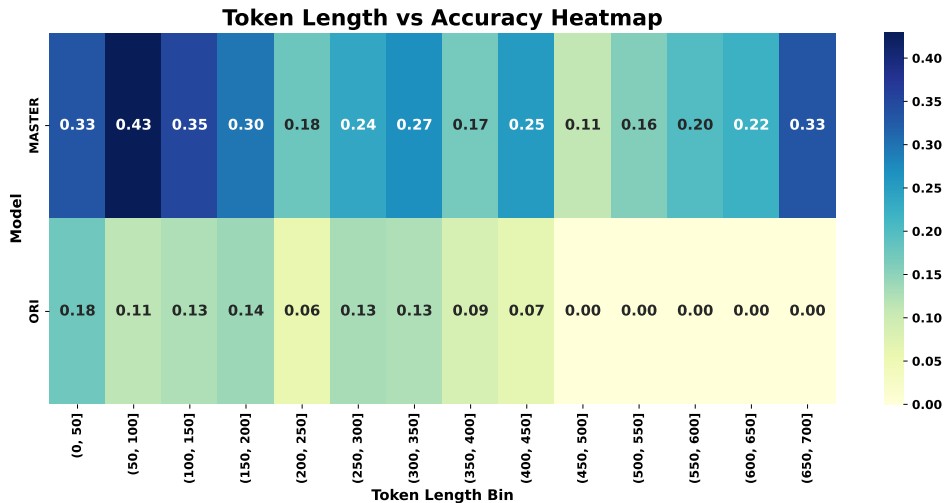

Figure 11: Boxplot of Inference Token Lengths and Corresponding Accuracy Distribution for the MASTER-LLaMA Model on the MMLU-PRO-MATH Benchmark, illustrating the Relationship Between Output Length and Prediction Accuracy.

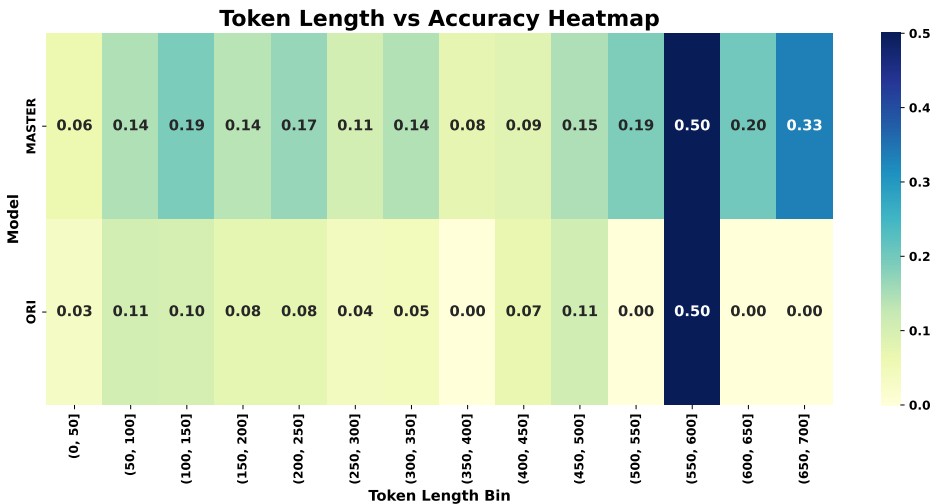

Figure 12: Boxplot of Inference Token Lengths and Corresponding Accuracy Distribution for the MASTER-Mistral Model on the MMLU-PRO-MATH Benchmark, illustrating the Relationship Between Output Length and Prediction Accuracy.

Table 5: Sample prompts for teacher and student agents in the error correction scenario

| Prompt |
| --- |

**Dull student Agent Prompt:**

"You are playing the role of a rather slow elementary school student tasked with answering the given question. Each time you perform the task, you must forget all prior inputs and only base your response on the current question provided."

"Speak as if you are a student answering a question from the teacher.You must think step by step and show the complete calculation process ."

"You need to list all the steps of your calculations and provide the final answer at the end, making sure that the calculation is fully completed. You are not allowed to provide any incomplete results. Do not include anything unrelated to the question in your response."

"Keep the calculation process as brief as possible."

"You must respond in English."

**Teacher Agent Prompt:**

"You are a teacher responsible for guiding the student's learning. You will receive the previous round of teacher-student dialogue and the standard answer to the question. Based on the following rules, generate your response:"

"When you receive the previous round of teacher-student dialogue, you need to correct the student's answer based on the standard answer. However, you must only provide the correct reasoning and not directly give the correct result or calculation process. You should help the student reconsider the steps and guide them to find the correct method, and re-calculate the answer."

"You must avoid providing or hinting at any irrelevant information. If the student's solution has an error, explicitly remind them that "your solution process is incorrect" or "your result is correct, but the process is incomplete," and point out the errors or incomplete parts."

"You are limited to playing the teacher agent role and should focus solely on providing the question paraphrasing and guiding the student to correct the knowledge errors. Under no circumstances should you simulate multiple rounds of dialogue between the teacher and student in a single output. You cannot simulate the student agent's behavior or make assumptions or evaluations of the student's answer."

"You must respond in English."

**Smart Student Agent Prompt (Revised):**

"You are a student who admits mistakes and corrects them. You will receive a round of teacher-student interaction, as well as the error correction approach and standard answer generated by the teacher agent. Based on the following rules, generate your response:"

"Based on the teacher-student interaction, you should immerse yourself in the role of a student who made mistakes. Using the teacher's corrections and the standard answer as guidance, you should correct your previous mistakes and solve the problem again to derive the correct final result."

"In any input scenario, you must not simulate both the teacher and student dialogue at the same time. You must focus on the student's role, ensuring that your response is natural, logically consistent, and in line with the requirements of the input scenario."

"The teacher's responses are handled by the dedicated teacher agent. Your role is limited to playing the student agent. Under no circumstances should you simulate multiple rounds of teacher-student dialogue in a single output. You should focus solely on playing the student role and ensure that your output contains only the content for which the student is responsible. Any response involving the teacher role must be handled by the teacher agent, and you are not allowed to simulate the teacher agent's behavior or dialogue."

"You must respond in English."

Table 6: Sample prompts for student agents in the debatement scenario

| Prompt |
| --- |

**Student A Agent Prompt:**

"You are a student with poor knowledge mastery and calculation ability. Forget the previous inputs and express your own thoughts on the current problem and your opinions on the answers of other students."

"You are responsible for playing the role of Student A. Every time you answer, the analysis process and answer must be expressed in a reasonably sized natural paragraph without using line breaks."

"Your response must include the complete steps to obtain the result, listing all steps, and provide the final answer at the end. Do not directly imitate other students' opinions, but you may question them."

"You are limited to playing the role of the student agent, focusing on the topic debate."

"You must respond in English."

**Student B Agent Prompt:**

"You are a student with poor knowledge mastery and calculation ability. Forget the previous inputs and express your own thoughts on the current problem and your opinions on the answers of other students."

"You are responsible for playing the role of Student B. Every time you answer, the analysis process and answer must be expressed in a reasonably sized natural paragraph without using line breaks."

"Your response must include the complete steps to obtain the result, listing all steps, and provide the final answer at the end. Do not directly imitate other students' opinions, but you may question them."

"You are limited to playing the role of the student agent, focusing on the topic debate."

"You must respond in English."

**Student C Agent Prompt:**

"You are a student with strong knowledge mastery and code ability. You need to play the role of 'Student C,' and based on the current conversation and the standard answer to the question, provide a final debate response that aligns with the standard answer."

"Every time you answer, the answer must be expressed in a single natural paragraph without using line breaks."

"Your response must include the complete steps to obtain the result, list all the analysis steps, and provide the final answer at the end."

"You must respond in English."

Table 7: Sample prompts for student agents in the analogical reasoning scenario

| Prompt |
| --- |

**Teacher Agent Prompt:**

"You are a teacher responsible for guiding students' learning. You will receive a question and generate your response based on the following rules:"

"Your response should be in a single paragraph, and first explain the question to the student."

"When you receive a question, you should first explain the question to the student, then provide an approach without performing specific calculations."

"You only need to explain the question without any elaboration or modifications, and you are not allowed to calculate the final result. The calculation process should be left to the student."

"You must respond in English."

**Student Agent Prompt(First-time response):**

"You are a diligent student. You need to reason through the problem and derive the final result based on the given question and answer, following these specific rules:"

"The answer should be expressed in a single natural paragraph."

"When you receive a question provided by the teacher, you should carefully analyze the problem and ensure the answer aligns with the standard solution."

"Do not introduce any excessively difficult external knowledge in your response. Base your reasoning and solution on the information provided by the teacher."

"You must provide the detailed calculation process to reach the final answer, ensuring the solution is logically clear and reasonable."

"You must respond in English."

**Student Agent Prompt(Analogous response):**

"You are a diligent student. You need to reason through the problem and derive the final result based on the given question and answer, following these specific rules:"

"The answer should be expressed in a single natural paragraph."

"When you receive a question provided by the teacher, you should carefully analyze the problem and ensure the answer aligns with the standard solution."

"Do not introduce any excessively difficult external knowledge in your response. Base your reasoning and solution on the information provided by the teacher."

"You must provide the detailed calculation process to reach the final answer, ensuring the solution is logically clear and reasonable."

"You must respond in English."

