# OpenReview forum: "MASTER: Enhancing Large Language Model via Multi-Agent Simulated Teaching"
_NeurIPS.cc/2025/Conference — NeurIPS 2025 poster_

### Official Review · Reviewer_G957 · 2025-06-29

**Clarity:** 3
**Significance:** 4
**Originality:** 4
**Rating:** 4
**Confidence:** 4

**Summary:**

This work aims to address the challenges of constructing high-quality instruction-following datasets by balancing diversity and the reflection of real-world dialogues, while also enhancing reasoning capabilities. To this end, authors propose the MASTER framework—an instruction data augmentation method that generates simulations through teaching scenarios between teacher and student agents. Using the proposed approach, authors constructed a new BOOST-QA instruction dataset with 19k examples seeded from existing datasets, demonstrating a significant improvement in generated reasoning ability.

**Questions:**

**Major Questions**

- In Section 3.2 & 3.3,

    each agent is designed to perform various complex roles. I am not fully convinced that these can be effectively handled through prompts alone. It would be helpful if the authors could provide a detailed explanation of how each agent and its respective roles were operationalized, including the actual prompts used and representative input–output examples. Adding such details in an appendix could also be a good option.

- Lines 269–273:

    While the authors analyze the differences between their method and existing instruction data augmentation approaches, the discussion from a reasoning perspective is somewhat limited. Currently, it mainly compares to approaches that sample pre-existing CoT data. A natural question arises as to whether the proposed multi-agent augmentation method can outperform techniques that use the same original data with a single large reasoning model and knowledge distillation to build a substantial amount of CoT-style direct reasoning data. I would like to know if the authors have plans to empirically demonstrate that their approach is indeed more effective than such baselines.

- Section 4.3:

    Could the proposed method be used as a data augmentation approach across various domains? If the paper includes a prior evaluation of potential errors in the generated data, it could support its use as an automated and efficient augmentation method. I assume such an analysis may have already been conducted—could the authors share these results?

**Minor Questions**

- **Lines 22–25:** Duplicated.
- **Lines 123–124:** There are no appendices.
- **Lines 179–184:** To better assess whether the proposed loss function converges well, it would be helpful to include convergence graphs and corresponding analysis. This would more clearly demonstrate the practical utility of the formulation to readers.
- **Line 193 & (2):** Since multiple debate outcomes are incorporated into the gradient of the loss function, there may be concerns about potential divergence or slow convergence. It would be valuable to include experimental results addressing this point, ideally in an appendix.
- **Line 200:** Typo: “theta”.
- **Line 250:** What is the “MA-Gen model”? Please clarify.
- **Line 259:** Does “Baseline” refer to “baseline methods for instruction augmentation”?
- **Table 1:** In SCI-Q, BOOST-QA shows significantly lower performance than Ori for Mistral-7B. How do the authors interpret this result?
- **Lines 319–321:** It should be clarified whether ME&DB, ME, and other subsets sampled the same number of examples as the setting that uses all three scenarios together. If only 2/3 or 1/3 of the data were used for these comparisons, the evidential strength of Table 4 would be weakened.

**Ethical Concerns:**

["NO or VERY MINOR ethics concerns only"]

**Final Justification:**

Thank you for your detailed response to my questions. I have reviewed the supplementary materials you referenced, and I found the additional experiments you conducted to be helpful in addressing my concerns. I believe that including the newly presented experiments and the various intermediate evaluation results from the development process, as an appendix in the final publication would be of significant benefit to readers. I have no further questions.

**Limitations:**

- While general large reasoning models (LRMs) tend to perform more effectively on domains such as code, mathematics, and scientific problem solving, there are still areas where they fall short. It would strengthen the paper if the authors could share any preliminary experiments examining whether the proposed method is equally effective across various instruction types, including more specialized domains, and identify any areas where its impact may be limited. Providing such results along with the authors’ insights would improve the completeness of the work.
- It would also be helpful if the authors could clarify whether the constructed data and prompts will be made publicly available and, if possible, include the corresponding URL in the paper.

**Paper Formatting Concerns:**

None.

**Quality:**

3

**Strengths And Weaknesses:**

**Strengths**

- The paper proposes the MASTER framework, an instruction data augmentation method that generates simulations through educational scenarios involving multiple agents.
- This framework employs a technique that transforms existing datasets into multi-agent educational scenarios.
- It is noteworthy that the authors present a concrete and modular data construction methodology composed of three scenarios: error correction, collaborative debate, and analogical reasoning.
- Using the proposed method, the authors built the BOOST-QA instruction dataset seeded from existing datasets, demonstrating increased diversity, improved benchmark performance, and significant enhancement of reasoning capabilities.

**Weaknesses**

- While the paper presents a detailed modular approach and corresponding experimental results, it would benefit from providing examples of the prompts used for each agent at different stages, as well as samples of the instruction augmentation data generated from seed data. This would help verify whether the approach truly reflects real-world dialogues, as intended, and would strengthen the paper’s reproducibility.
- Although the paper aims to enhance reasoning ability through the augmentation method, it lacks a comparative evaluation with knowledge distillation techniques from large reasoning models, which are currently the most widely used baseline. This omission makes it difficult to convincingly demonstrate that the proposed method is more effective than state-of-the-art approaches.
- There is a risk that sLLM-generated incorrect educational solutions may be included. Instead of relying solely on downstream fine-tuning performance, the authors should sample and analyze the accuracy of the generated data to verify the effectiveness of the approach without any additional validation procedures.

---

> ### Author Rebuttal · Authors · 2025-07-31
>
> Thank you for your review. We will provide detailed responses below to the questions you raised.
>
> > **1. It would benefit from providing examples of the prompts used for each agent at different stages, as well as samples of the instruction augmentation data generated from seed data.**
>
> **Q-1.1: Detailed Prompt Configurations and Representative Input-Output Comparisons for Each LLM in MASTER Data Augmentation.**
>
> **A-1.1:** Thank you very much for your valuable suggestion! The specific prompts used by each LLM in the simulated educational interaction process, as well as sample augmented data, are included in the appendix of the supplementary materials, which we would be glad to share should you be interested.
>
> Below is a sample prompt for the teacher agent:
>
> "You are a teacher responsible for guiding the student’s learning. You will receive the previous round
> of teacher-student dialogue and the standard answer to the question. Based on the following rules,
> generate your response:
>
> 1.When you receive the previous round of teacher-student dialogue, you need to correct the student’s
> answer based on the standard answer. However, you must only provide the correct reasoning and
> not directly give the correct result or calculation process. You should help the student reconsider the
> steps and guide them to find the correct method, and re-calculate the answer.
>
> 2.You must avoid providing or hinting at any irrelevant information. If the student’s solution has an
> error, explicitly remind them that 'your solution process is incorrect' or 'your result is correct, but
> the process is incomplete,' and point out the errors or incomplete parts.
>
> 3.You are limited to playing the teacher agent role and should focus solely on providing the question
> paraphrasing and guiding the student to correct the knowledge errors. Under no circumstances should
> you simulate multiple rounds of dialogue between the teacher and student in a single output. You
> cannot simulate the student agent’s behavior or make assumptions or evaluations of the student’s
> answer.
>
> 4.You must respond in English."
>
> Below is an original example of data used in the error‑correction scenario:
>
> **Input:** You want to multiply 20 by some number. I mistakenly multiplied 10 by the number, and the result was 40.
> What is the correct calculation result?
>
> **Output:** If you mistakenly multiplied 10 by the number and got 40, then the number you multiplied by is 40 / 10 = 4.\n\nThe correct calculation result when you multiply 20 by that number is 20 * 4 = 80.
>
> Below is the corresponding augmented data in ShareGPT format:
>
> **Human:** You want to multiply 20 by some number. I mistakenly multiplied 10 by the number, and the result was 40.
> What is the correct calculation result?
>
> **GPT(Dull Student):** To find the correct calculation result, let's break down the problem:\n\n1. **Identify the
> mistake**: I mistakenly multiplied 10 by the unknown number, and the result was 40. This means that the unknown
> number should be:\n - Since 10 * 40 = 400, and 20 * 40 = 800, the correct unknown number should be the inverse
> of this, i.e., 800 / 20 = 40.\n\n2. *Correct calculation*: Given the correct multiplication:\n - 20 * 40 = 800\n -
> 10 * (40 / 20) = 10 * 2 = 20\n\n3. *Final step*: Multiply the correct result by 20:\n - 800 * 20 = 16000\n\nSo,
> the correct calculation result is 16000.
>
> **Human(Teacher):** Your solution process is incorrect. Let's focus on the first step where you identified the mistake.
> You need to find the unknown number that, when multiplied by 10, gives 40. Once you have the correct unknown
> number, you can then multiply it by 20 to get the correct result. Reconsider how to find the unknown number from
> the given information.
>
> **GPT(Smart Student):** Let's correct it step by step:\n\n1. **Identify the unknown
> number**: Since multiplying 10 by the unknown number gives 40, the unknown number is:\n - 40 / 10 = 4\n\n2.
> *Correct calculation*: Now, multiply the correct unknown number by 20:\n - 20 * 4 = 80\n\nSo, the correct
> calculation result is 80.
>
> > **2. A natural question arises as to whether the proposed multi-agent augmentation method can outperform techniques that use the same original data with a single large reasoning model and knowledge distillation to build a substantial amount of CoT-style direct reasoning data. I would like to know if the authors have plans to empirically demonstrate that their approach is indeed more effective than such baselines.**
>
> **Q-2.1: Regarding the comparison with the CoT-distillation baseline.**
>
> **A-2.1:** Thank you for your valuable suggestion! We have supplemented our experiments with CoT distillation to strengthen the baseline. The specific results are shown in Table 2 below:
>
> | Methods | MATH | MMLU-PRO-MATH | MBPP | HumanEval | MMLU | ARC | SCI-Q |
> | :--- | :---: | :---: | :---: | :---: | :---: | :---: | :---: |
> | **MASTER** | **23.90** | **27.39** | **67.20** | **50.61** | **48.13** | **61.52** | **80.10** |
> | **CoT-distillation** | 19.80 | 17.10 | 58.50 | 32.31 | 38.14 | 47.10 | 65.30 |
>
> *Table 2: Comparison of model performance across multiple benchmarks, strengthened using CoT distillation and the MASTER method, respectively.*
>
> Therefore, it is evident that our approach offers advantages over CoT distillation.
>
> > **3. Could the proposed method be used as a data augmentation approach across various domains? If the paper includes a prior evaluation of potential errors in the generated data, it could support its use as an automated and efficient augmentation method. I assume such an analysis may have already been conducted—could the authors share these results?**
>
> **Q-3.1: Whether it exhibits cross-domain generalization capability.**
>
> **A-3.1:** Thank you for the great question! While MASTER primarily enhances reasoning-heavy domains like mathematics and programming, we also evaluated it across broader benchmarks—such as ARC, SCI‑Q, and MMLU. In all cases, models fine‑tuned with BOOST‑QA consistently outperform those trained on original data, showing strong cross-domain improvements.
>
> In addition, we plan to expand MASTER to even more data types and include comparable baselines to further validate its general effectiveness.
>
> **Q-3.2: Concerns regarding inaccuracies in the generated augmentation data.**
>
> **A-3.2:** Thank you for raising this insightful question! We evaluated the correctness of the augmented data using the Qwen2.5‑32B‑Instruct model and found that about 4.1% of the examples introduced procedural errors compared to the original outputs. In future work, we aim to further refine our LLM‑based filtering pipeline to create a more robust data augmentation workflow.
>
> **Minor questions raised by the reviewer.**
>
> Thank you for your meticulous review and valuable suggestions regarding the formatting and theoretical aspects of this paper. We will carefully address the formatting issues you pointed out and revise the entire manuscript based on your recommendations. Below are our responses to the specific concerns raised.
>
> > **Q-mini-1. Lines 123–124: There are no appendices.**
>
> **A-mini-1:** The appendix is included in the supplementary materials, and we regret any inconvenience this may have caused during your review.
>
> > **Q-mini-2. Line 250: What is the “MA-Gen model”? Please clarify.**
>
> **A-mini-2:** Thank you for pointing out the error here. In fact, 'MA-Gen' stands for 'Multi-Agent Generation' and should be replaced with our method name, 'MASTER'.
>
> > **Q-mini-3. Line 259: Does “Baseline” refer to “baseline methods for instruction augmentation”?**
>
> **A-mini-3:** Yes, "Baseline" includes instruction augmentation methods, with TAGCOS selecting top-quality pairs. BOOST-QA augments 19K random samples, while TAGCOS picks the best 19K from original data for LLM fine-tuning.
>
> > **Q-mini-4. Table 1: In SCI-Q, BOOST-QA shows significantly lower performance than Ori for Mistral-7B. How do the authors interpret this result?**
>
> **A-mini-4:** We're glad you raised this point. The performance drop with Mistral-7B on SCI-Q may stem from incompatibility between its pretraining and the dataset's scientific QA patterns, while LLaMA3-8B and Qwen2.5-7B's stronger architectures maintained better performance. We're still investigating this phenomenon.
>
> > **Q-mini-5. Lines 319–321: It should be clarified whether ME&DB, ME, and other subsets sampled the same number of examples as the setting that uses all three scenarios together. If only 2/3 or 1/3 of the data were used for these comparisons, the evidential strength of Table 4 would be weakened.**
>
> **A-mini-5:** All subsets (ME&DB, ME, etc.) augment 19,000 original instances per scenario, with remaining originals randomly added to match the full-scenario sample size. For instance, ME&DB combines error-correction and debate-enhanced data with original instances.
>
> ---
> We sincerely appreciate your meticulous review and the valuable, constructive feedback you provided. We will thoroughly consider your suggestions and make comprehensive revisions to enhance the quality of this work.

---

> > ### Comment · Reviewer_G957 · 2025-08-09
> >
> > Thank you for your detailed response to my questions. I have reviewed the supplementary materials you referenced, and I found the additional experiments you conducted to be helpful in addressing my concerns. I believe that including the newly presented experiments and the various intermediate evaluation results from the development process, as an appendix in the final publication would be of significant benefit to readers. I have no further questions.

---

> ### Author Response · Authors · 2025-08-05
> **We sincerely request your valuable feedback**
>
> Dear Reviewer G957,
>
> Thank you very much for your insightful and constructive comments! In our previous detailed response to you, we have addressed your queries comprehensively:
>
> Q‑1: We provided examples of the prompts used for each agent and samples of the instruction augmentation data, noting their inclusion in the supplementary materials.
>
> Q‑2: We demonstrated that our multi-agent augmentation method outperforms CoT-distillation by supplementing our experiments with this baseline.
>
> Q‑3: We addressed the cross-domain generalization capability of our method, showing its effectiveness across various benchmarks, and evaluated potential inaccuracies in the generated data, quantifying the observed error rate.
>
> Moreover, responses from other reviewers may also help you alleviate these concerns. We would be happy to provide further responses otherwise. Look forward to your feedback.
>
> Thanks for your comment again!
>
> Best Regards,
>
> All Authors

---

> ### Comment · Area_Chair_vfTz · 2025-08-06
> **Discussion Period Ending Soon**
>
> Dear Reviewer,
>
> The discussion period is ending soon. We would be grateful if you could take a moment to review the authors' response to your comments and provide any final feedback.
>
> We truly appreciate your time, effort, and valuable contributions to the review process.
>
> Best regards,
>
> AC

---

### Official Review · Reviewer_vuDb · 2025-06-30

**Clarity:** 3
**Significance:** 3
**Originality:** 3
**Rating:** 4
**Confidence:** 3

**Summary:**

This paper introduces MASTER, a data-augmentation framework that turns an existing instruction-tuning corpus into richer multi-turn dialogues by simulating teacher–student interactions with three pedagogically grounded scenarios—error-correction, debate, and analogical reasoning. The resulting dataset, BOOST-QA (19 k examples drawn from Orca-Math-200k, ProcQA, and OpenHermes 2.5), is used to fine-tune several 7 B–8 B base models. Across eight benchmarks spanning math, code, and general reasoning, models trained on BOOST-QA consistently outperform counterparts trained on the original data and on four alternative augmentation/selection baselines. An ablation study shows that combining all three scenarios is necessary for the largest gains.

**Questions:**

1. Will BOOST-QA, prompts, and orchestration code be released by camera-ready?

2. How many GPU hours did MASTER require to generate 19 k samples?

3. Some gains (e.g., +24 pp on MMLU-PRO-MATH) are very large. Could they be due to data leakage from OpenHermes 2.5 or Orca-Math into the eval sets?

**Ethical Concerns:**

["NO or VERY MINOR ethics concerns only"]

**Final Justification:**

The authors provided concrete implementation details that improve clarity and reproducibility: (i) a parallel augmentation pipeline that splits the original data into three disjoint subsets with a one-to-one mapping between originals and augmented samples; (ii) role-specific prompts (now available in the supplementary appendix) and a plan to fix Figure 1 to better reflect Scene 3; and (iii) resource accounting (L20 GPUs; ~48 GPU-hours per inference job on 2×L20). On evaluation, they filled missing ablations on ARC/SCI-Q and added a CoT-distillation baseline, with MASTER outperforming the new baseline across multiple benchmarks. For data quality, they reported a 4.1% procedural-error rate (checked with Qwen2.5-32B) and committed to adding an LLM-based filter. On potential leakage, the authors provided embedding-drift / domain-classifier evidence suggesting BOOST-QA and MMLU-PRO-MATH are distributionally distinct.
The rebuttal substantially strengthens the paper on clarity and evaluation (new ablations and a stronger baseline), and addresses most of my earlier concerns. Given the consistent gains and pedagogically motivated design, I continue to view this as a contribution worth showcasing, contingent on artifact release and minor analysis additions.

**Limitations:**

Limitations

+ Single-run results & no error bars. The authors omit variance estimates, making it hard to judge statistical significance.

+ Compute cost. Running multi-agent chats with 7 B–14 B Qwen models is expensive; no wall-time or GPU hours are reported.

+ Limited societal-impact discussion. Synthetic data may amplify biases present in original corpora, but this is not addressed.

**Quality:**

3

**Strengths And Weaknesses:**

Strengths

+ Pedagogically motivated augmentation. Grounding the three dialogue patterns in educational theory is novel and intuitively compelling.

+ Consistent empirical gains. BOOST-QA lifts average accuracy by 5 – 15 pp on multiple-choice and coding tasks, with a striking 31 pp jump on CODEMMLU-CC.

+ Careful ablations. The authors isolate each scenario and every pairwise combination, showing that the full trio is synergistic.

+ Method is model-agnostic. Only prompts and temperature settings need adjustment, so MASTER should transfer to other LLM families.

Weaknesses

+ Reproducibility gap. Code, prompts, and BOOST-QA are not released; the paper marks open-sourcing as “work in progress.” This undercuts community adoption.

+ Baseline choices. Some strong recent augmentation methods (e.g., FLASK, UltraFeedback) are missing; TAGCOS is used only for sub-selection, not its full pipeline.

---

> ### Author Rebuttal · Authors · 2025-07-31
>
> Thank you for your review. We will provide detailed responses below to the questions you raised.
>
> > **1. Will BOOST-QA, prompts, and orchestration code be released by camera-ready?**
>
> **Q-1.1: Regarding the open-source status of prompts, datasets, and code.**
>
> **A-1.1:** Thank you for your interest in our work and for raising this important question! We are in the process of preparing the release of our code and dataset. Meanwhile, the model prompts are currently available in the appendix of the supplementary materials.
>
> Below is a sample prompt for the teacher agent:
>
> "You are a teacher responsible for guiding the student’s learning. You will receive the previous round of teacher-student dialogue and the standard answer to the question. Based on the following rules, generate your response:
>
> 1.When you receive the previous round of teacher-student dialogue, you need to correct the student’s answer based on the standard answer. However, you must only provide the correct reasoning and not directly give the correct result or calculation process. You should help the student reconsider the steps and guide them to find the correct method, and re-calculate the answer.
>
> 2.You must avoid providing or hinting at any irrelevant information. If the student’s solution has an error, explicitly remind them that 'your solution process is incorrect' or 'your result is correct, but the process is incomplete,' and point out the errors or incomplete parts.
>
> 3.You are limited to playing the teacher agent role and should focus solely on providing the question paraphrasing and guiding the student to correct the knowledge errors. Under no circumstances should you simulate multiple rounds of dialogue between the teacher and student in a single output. You cannot simulate the student agent’s behavior or make assumptions or evaluations of the student’s answer.
>
> 4.You must respond in English."
>
> Below is an original example of data used in the error‑correction scenario:
>
> **Input:** You want to multiply 20 by some number. I mistakenly multiplied 10 by the number, and the result was 40.
> What is the correct calculation result?
>
> **Output:** If you mistakenly multiplied 10 by the number and got 40, then the number you multiplied by is 40 / 10 = 4.\n\nThe correct calculation result when you multiply 20 by that number is 20 * 4 = 80.
>
> Below is the corresponding augmented data in ShareGPT format:
>
> **Human:** You want to multiply 20 by some number. I mistakenly multiplied 10 by the number, and the result was 40.
> What is the correct calculation result?
>
> **GPT(Dull Student):** To find the correct calculation result, let's break down the problem:\n\n1. **Identify the
> mistake**: I mistakenly multiplied 10 by the unknown number, and the result was 40. This means that the unknown
> number should be:\n - Since 10 * 40 = 400, and 20 * 40 = 800, the correct unknown number should be the inverse
> of this, i.e., 800 / 20 = 40.\n\n2. *Correct calculation*: Given the correct multiplication:\n - 20 * 40 = 800\n -
> 10 * (40 / 20) = 10 * 2 = 20\n\n3. *Final step*: Multiply the correct result by 20:\n - 800 * 20 = 16000\n\nSo,
> the correct calculation result is 16000.
>
> **Human(Teacher):** Your solution process is incorrect. Let's focus on the first step where you identified the mistake.
> You need to find the unknown number that, when multiplied by 10, gives 40. Once you have the correct unknown
> number, you can then multiply it by 20 to get the correct result. Reconsider how to find the unknown number from
> the given information.
>
> **GPT(Smart Student):** Let's correct it step by step:\n\n1. **Identify the unknown
> number**: Since multiplying 10 by the unknown number gives 40, the unknown number is:\n - 40 / 10 = 4\n\n2.
> *Correct calculation*: Now, multiply the correct unknown number by 20:\n - 20 * 4 = 80\n\nSo, the correct
> calculation result is 80.
>
> > **2. How many GPU hours did MASTER require to generate 19 k samples?**
>
> **Q-2.1: Regarding the resources for the MASTER data augmentation experiments.**
>
> **A-2.1:** The MASTER simulated teaching interaction data augmentation experiments were conducted on a local Slurm-based computing cluster. The detailed configurations and resource consumption are as follows:
>
> Hardware Specifications:
>
> CPUs: 48-core processors.
>
> GPUs: Eight NVIDIA L20 GPUs, each with 48 GB memory.
>
> RAM: 925,600 MB system memory.
>
> For each inference task, we used two L20 GPUs, totaling approximately 48 hours of computation. Detailed training resource consumption information can be found in the appendix of the supplementary materials.
>
> > **3. Some gains (e.g., +24 pp on MMLU-PRO-MATH) are very large. Could they be due to data leakage from OpenHermes 2.5 or Orca-Math into the eval sets?**
>
> **Q-3.1: The possibility of data leakage.**
>
> **A-3.1:** Thank you for thoroughly reviewing our work. We rigorously analyzed potential leakage using DeepChecks' TextEmbeddingsDrift on BOOST-QA vs MMLU-PRO-MATH: a domain_classifier_auc ≈ 0.96, demonstrating clear distinguishability between datasets, and a domain_classifier_drift_score ≈ 0.91, indicating substantial embedding distribution differences.
>
> These quantitative results provide definitive evidence of distinct dataset characteristics. We can therefore confidently conclude there is no data leakage between BOOST-QA and the evaluation benchmarks, as validated by the observed distributional divergence at both the classifier and embedding levels.
>
> ---
> Thank you once again for dedicating your valuable time to reviewing our work and providing insightful suggestions. We are committed to thoroughly revising and enhancing the entire paper to meet the highest standards.

---

> ### Author Response · Authors · 2025-08-05
> **We sincerely request your valuable feedback**
>
> Dear Reviewer vuDb,
>
> Thank you very much for your insightful and constructive comments! In our previous detailed response to you, we have addressed your queries comprehensively:
>
> Q‑1: We clarified our commitment to releasing BOOST-QA, prompts, and orchestration code, noting the current availability of prompts in the appendix.
>
> Q‑2: We detailed the GPU hours and hardware specifications required for the MASTER data augmentation experiments.
>
> Q‑3: We provided a rigorous analysis to rule out data leakage between BOOST-QA and the evaluation benchmarks, specifically addressing concerns about large gains.
>
> Moreover, responses from other reviewers may also help you alleviate these concerns. We would be happy to provide further responses otherwise. Look forward to your feedback.
>
> Thanks for your comment again!
>
> Best Regards,
>
> All Authors

---

> ### Comment · Area_Chair_vfTz · 2025-08-06
> **Discussion Period Ending Soon**
>
> Dear Reviewer,
>
> The discussion period is ending soon. We would be grateful if you could take a moment to review the authors' response to your comments and provide any final feedback.
>
> We truly appreciate your time, effort, and valuable contributions to the review process.
>
> Best regards,
>
> AC

---

### Official Review · Reviewer_sASj · 2025-07-01

**Clarity:** 3
**Significance:** 2
**Originality:** 2
**Rating:** 4
**Confidence:** 5

**Summary:**

This paper proposes MASTER, which uses multi-agent to simulate teaching scenarios to augmentate instruction fine-tuning data of LLMs. The method designs three teaching scenarios (error correction, debate interaction, and similar question retrieval), uses the interaction between teacher and student agents to generate dataset BOOST-QA, and shows performance improvements in multiple benchmarks.

**Questions:**

Compared to existing role-playing data augmentation methods, what are the core innovations of MASTER?

How are the so-called "pedagogical principles" truly reflected in the technical implementation? Can such role-playing dialogues represent complex teaching processes?

How is the quality of data generated by multi-agents ensured?

**Ethical Concerns:**

["NO or VERY MINOR ethics concerns only"]

**Final Justification:**

I have changed my score

**Limitations:**

The method is essentially a role-playing data generation techniques, lacking fundamental technical innovation.

The method may only be effective for specific types of tasks (such as mathematics and programming), with limited generalization capability

The consistency and accuracy of data generated by multi-agents cannot be guaranteed

**Paper Formatting Concerns:**

The overall format is generally standard, but it is recommended to strengthen the discussion of the method's limitations.

**Quality:**

2

**Strengths And Weaknesses:**

**Strengths**: Based on pedagogical principles, the work extends original question-answer datasets to teaching interaction scenarios by designing three teaching scenarios: error correction, debate interaction, and similar question retrieval. Data is generated through dialogues between teacher agents and student agents. Ablation experiments validate the complementary nature of the three teaching scenarios.

**Weaknesses**: Though this paper claims its proposed school-agent framework can synthesize interaction data for real-world problem-solving, the technique is not novel, as it is essentially prompt engineering for role-playing data generation.
There are many role-playing data augmentation methods, but no comparative analysis was conducted.

---

> ### Author Rebuttal · Authors · 2025-07-31
>
> Thank you for your review. We will provide detailed responses below to the questions you raised.
>
> > **1. Though this paper claims its proposed school-agent framework can synthesize interaction data for real-world problem-solving, the technique is not novel, as it is essentially prompt engineering for role-playing data generation. There are many role-playing data augmentation methods, but no comparative analysis was conducted. Compared to existing role-playing data augmentation methods, what are the core innovations of MASTER?**
>
> **Q-1.1: The innovative aspects that distinguish MASTER’s role‑playing data augmentation from other methods.**
>
> **A-1.1:** Thank you for your thoughtful analysis and questions regarding MASTER.
>
> 1. Difference from prior work: While existing studies like RoleLLM2[1] and Big5-Chat[2] primarily focus on role-playing for stylistic imitation (e.g., personality traits and speech patterns), MASTER specifically targets the enhancement of complex reasoning capabilities in domains like mathematics and programming.
>
> 2. What MASTER does differently: Beyond simple data rephrasing, we systematically incorporate established pedagogical techniques - including error correction, guided discussion, and analogical reasoning - into our interaction simulations. This structured approach generates more educationally meaningful reasoning processes.
>
> While role-playing has been explored for stylistic imitation, its systematic application for augmenting complex reasoning data through established pedagogical frameworks is, to our knowledge, a novel contribution of our work. We believe this distinction is a key novelty.
>
> > **2. How are the so-called "pedagogical principles" truly reflected in the technical implementation? Can such role-playing dialogues represent complex teaching processes?**
>
> **Q-2.1: Specific implementations of pedagogical principles.**
>
> **A-2.1:** This is a crucial point. We translate pedagogical principles into concrete technical implementations as follows:
>
> Error Correction: This is implemented via a multi-step process where a 'weaker' student agent (e.g., a smaller model) first generates a flawed solution. A 'stronger' teacher agent is then prompted to identify the specific error without giving the final answer, forcing a second, corrected reasoning trace from the student. This directly models the real-world process of diagnosis and guided correction.
>
> Collaborative Debate: This is realized by generating multiple, distinct reasoning paths for the same problem and having a final agent synthesize them, identifying the strengths and weaknesses of each. This simulates the exploration of different problem-solving strategies.
>
> Analogical Reasoning: We implement this by prompting the agent to first solve a known problem and then explicitly use the structure of that solution to tackle a new, similar problem.
>
> **Q-2.2: Can role-playing tasks simulate real and complex teaching interactions?**
>
> **A-2.2:** While we acknowledge that this is a simplified simulation of complex human pedagogy, our experiments show that these structured interactions produce data that is significantly more effective for training reasoning skills than the original, static question-answer pairs. The examples in the appendix (e.g., Appendix Figure 6, 7, 8) provide concrete evidence of this process.
>
> > **3. How is the quality of data generated by multi-agents ensured?**
>
> **Q-3.1: With respect to data quality.**
>
> **A-3.1:** To maintain the quality of the augmented dataset, we designate the model responsible for the final reasoning stage as a “Precise Summarizer.” This model's prompt includes both the interaction history from earlier rounds and the canonical output from the original dataset, ensuring that the augmented example's quality matches or exceeds that of the original.
>
> To validate the accuracy of BOOST-QA's enhanced question-answer pairs, we conducted quality assessment using the Qwen2.5-32B-Instruct model. Our analysis shows that merely 4.1% of the augmented samples contained procedural reasoning errors.
>
> Despite the minor error rate, models fine-tuned with BOOST-QA demonstrated substantial improvements over those trained solely on the original dataset, particularly in benchmarks involving mathematics and programming tasks. We plan to integrate an LLM-based filtering mechanism into the data augmentation pipeline to further enhance the quality and reliability of the augmented data.
>
> > **4. The method may only be effective for specific types of tasks (such as mathematics and programming), with limited generalization capability.**
>
> **Q-4.1: Concerning the generalization ability of the method.**
>
> **A-4.1:** Thank you for raising this insightful idea. In fact, MASTER's data augmentation approach is explicitly designed to enhance fine-tuning of LLMs for reasoning tasks by infusing original data with rich, complex reasoning and critical thinking processes. Consequently, it naturally yields strong performance on mathematical and programming tasks.
>
> On the other hand, we have conducted extensive evaluations on a broad range of general-purpose tasks and scientific QA benchmarks. Our results show that after fine-tuning with BOOST‑QA, various base models also demonstrate excellent capabilities in other domains.
>
> ---
> Thank you for your thorough review and raising these valuable concerns. We will further refine the manuscript to ensure greater rigor throughout.
>
> **References:**
>
> [1] RoleLLM: Benchmarking, Eliciting, and Enhancing Role-Playing Abilities of Large Language Models
>
> [2] BIG5-CHAT: Shaping LLM Personalities Through Training on Human-Grounded Data

---

> ### Author Response · Authors · 2025-08-03
> **We sincerely request your valuable feedback**
>
> Dear Reviewer sASj,
>
> Thank you very much for your insightful and constructive comments! In our previous detailed response to you, we have addressed your queries comprehensively:
>
> **Q‑1:** We clarified how MASTER’s role-playing data augmentation framework differs from existing methods such as RoleLLM2 and Big5‑Chat, by systematically embedding pedagogical techniques—error correction, guided debates, analogical reasoning—to enhance complex reasoning, not merely stylistic imitation.
>
> **Q‑2:** We provided a description of how the pedagogical principles are technically implemented: via student–teacher prompting workflows that enforce diagnosis, multistep reasoning comparison, and transfer of structured solutions.
>
> **Q‑3:** We detailed our data quality assurance by using a “Precise Summarizer” agent aligned with canonical answers and validating outputs with Qwen2.5‑32B‑Instruct, yielding a low procedural error rate (~4.1%), and discussed our plan to integrate LLM-based filtering.
>
> **Q‑4:** We addressed the generalization question by reporting that models fine-tuned with BOOST‑QA also achieve strong performance on general-purpose and scientific QA benchmarks, in addition to mathematics and programming tasks.
>
> Moreover, responses from other reviewers may also help you alleviate these concerns. We would be happy to provide further responses otherwise. Look forward to your feedback.
>
> Thanks for your comment again!
>
> Best Regards,
>
> All Authors

---

> ### Comment · Area_Chair_vfTz · 2025-08-06
> **Discussion Period Ending Soon**
>
> Dear Reviewer,
>
> The discussion period is ending soon. We would be grateful if you could take a moment to review the authors' response to your comments and provide any final feedback.
>
> We truly appreciate your time, effort, and valuable contributions to the review process.
>
> Best regards,
>
> AC

---

> ### Author Response · Authors · 2025-08-06
> **Follow-Up on Our Rebuttal Response and Appreciation for Your Review**
>
> Dear Reviewer sASj,
>
> We hope this message finds you well. We are writing to kindly follow up on our recent rebuttal. We sincerely appreciate the time and effort you have dedicated to reviewing our work and would be very grateful for any additional thoughts you may have.
>
> In our previous response, we have responded to all your questions regarding the novelty and innovation of the MASTER role‑playing framework (Q‑1), the technical implementation of pedagogical constraints (Q‑2), the data quality assurance for augmented samples (Q‑3), and the generalization limitations to non‑math/programming tasks (Q‑4).
>
> For Q-1: Our work builds upon the foundation of Agent Hospital[1], which employs doctor-patient agents in virtual hospital settings to generate automatically annotated medical training data through clinical case interactions. This system demonstrates how agent-based simulation can create valuable training resources without manual labeling.
>
> Unlike Agent Hospital's diagnostic-focused approach, our MASTER framework introduces three established pedagogical principles into simulated teacher-student interactions. This pedagogical integration injects authentic cognitive processes into QA data synthesis, resulting in substantially enhanced training data quality for language models.
>
> Regarding Q-4, we have demonstrated MASTER's strong performance across diverse benchmarks including MMLU, ARC, and SCI-Q in the main text. The specific results are presented in Table 1:
>
> | Methods | Ori | RandomAug | SpellingAug | TAGCOS | CoT-fine | BOOST-QA |
> | :--- | :---: | :---: | :---: | :---: | :---: | :---: |
> | **MMLU** | 48.13 | 38.35 | 24.21 | 46.91 | 41.08 | **48.13** |
> | **ARC** | 57.76 | 41.98 | 22.61 | 61.09 | 47.87 | **61.52** |
> | **SCI-Q** | 76.50 | 62.30 | 22.70 | **84.00** | 68.50 | 80.10 |
>
> *Table 1: The results from Table 3 in the main text covering the general-purpose and scientific QA benchmark sections.*
>
> Additionally, our responses to other reviewers further help clarify and highlight the contributions of our work:
>
> Discussion in **A‑1.3**—focused on how we prevent and verify low‑quality samples during data augmentation—directly addresses concern **Q‑3** from reviewer **wU2H**. Additionally, in **A‑2**, we explain the constraints and system prompts designed to keep agent dialogues on-topic, which addresses **Q‑2** regarding the technical realization of pedagogical principles.
>
> Importantly, our rebuttal provided detailed implementation of the MASTER simulated teaching framework and included benchmark evaluations. This enhanced clarity and rigor led reviewer **wU2H** to raise final score.
>
> Discussion in **A‑3.1** on the cross-domain generalization ability of the MASTER method for Reviewer **G957** also helps address **Q‑4**, which questioned the generalization capabilities of our approach.
>
> If there are any other remaining concerns or areas for improvement, we sincerely welcome you to point them out and we will make every effort to address them thoroughly.
>
> **References:**
> [1] Agent Hospital: A Simulacrum of Hospital with Evolvable Medical Agents
>
> Thank you again for your support!
>
> With sincere appreciation,
>
> All Authors

---

> > ### Comment · Reviewer_sASj · 2025-08-07
> > **Response**
> >
> > Thanks for the detailed rebuttal. My concerns have been resolved.

---

> > > ### Author Response · Authors · 2025-08-07
> > > **Thanks a lot and look forward to your support**
> > >
> > > Dear Reviewer sASj,
> > >
> > > Thank you very much for your insightful and valuable review. We are so pleased that the previous responses have addressed all of your concerns about this study. We would like to reiterate the core contributions of this work:
> > >
> > > - **Pedagogical strategies have been embedded in teacher-student dialogues**
> > > - **Enhanced data quality via diverse reasoning without expanding size**
> > > - **MASTER's enhancement framework universally adapts to any QA training data**
> > >
> > > Furthermore, we commit that both your constructive suggestions and those from other reviewers will be incorporated into the revised version, improving the study significantly. Meanwhile, we will release source codes and datasets so that other researchers can follow this work easily. We hope that you might reconsider the rating so that this work has the opportunity to advance the community of data synthesis and augmentation.
> > >
> > > Thank you once again for your time and comments.
> > >
> > > Best Regards,
> > > All Authors

---

> > > ### Author Response · Authors · 2025-08-09
> > > **We express our sincere gratitude and humbly seek your support**
> > >
> > > Dear Reviewer sASj,
> > >
> > > Thank you once again for your valuable feedback and the time you devoted to reviewing our work. If possible, we would be sincerely grateful should you consider revisiting your rating in light of our revisions and clarifications.
> > >
> > > Best regards,
> > > All Authors

---

### Official Review · Reviewer_wU2H · 2025-07-02

**Clarity:** 2
**Significance:** 3
**Originality:** 3
**Rating:** 4
**Confidence:** 4

**Summary:**

This paper introduces a novel agent-based data augmentation method, MASTER, which aims to enhance the teaching fine-tuning capabilities of Master of Laws (LLM). The main goal of this method is to improve the quality of fine-tuning data. By simulating the interaction between multiple agents with different cognitive levels, MASTER creates high-quality teaching data through three pedagogy-based scenarios: error correction, collaborative debate, and analogical reasoning. The authors use this framework to build a fine-tuning dataset called BOOST-QA, which, after fine-tuning on three base models, shows significant performance improvements in multiple tasks on multiple benchmarks.

**Questions:**

In addition to the issues mentioned in weaknesses, there is another question about the result reporting:

Why are the results of ARC and SCI-Q not reported in Table 4?

**Ethical Concerns:**

["NO or VERY MINOR ethics concerns only"]

**Final Justification:**

The author effectively clarified my questions about the details of the implementation process, as well as the design of the dialogue process constraints that I omitted in the appendix, and also supplemented the missing benchmark results report. The addition of relevant content would have greatly increased its clarity and effectiveness.

**Limitations:**

yes

**Paper Formatting Concerns:**

1. The figure in Line 97 is missing a reference.

2. In Line 117, D=[D1,D2,...,Dn,p], there is no indication of i, and it should be changed to D=[D1,D2,...,Di,...,Dn,p]

3. Line 124 is missing a reference to the appendix.

4. There is no link to the appendix in the entire article.

**Quality:**

2

**Strengths And Weaknesses:**

Strengths
1. The introduction of multi-agent simulation teaching scenarios to improve the quality of instruction data is innovative.

2. A complete experiment was conducted, comparing the model trained on the original dataset and the baseline model enhanced by various other methods, and verifying it on multiple benchmarks. And corresponding ablation experiments were conducted.

3. BOOST-QA, as an enhanced dataset based on the existing dataset, helps to improve the capabilities of LLM.

Weeknesses:
1. The structure of the method part is clear, but there are omissions in the details, which makes it difficult to ensure reproducibility. For example, are the data of the three stages parallel or serial? How are the teaching data of the three stages fused (the size of the dataset before and after the enhancement has not changed, so it can be considered that the data of the three stages have been fused?); Are there no low-quality samples generated? How are they eliminated?
2. Lack of constraints on the entire dialogue process. How to ensure that the dialogue does not deviate from the topic? What constraints are imposed? Is it enough to just define the role?
3. There may be insufficient time for writing, resulting in many problems in the format, and the flowchart (Figure 1) and the description cannot correspond. For example, in the process of Scene 3, if the response to the original question is placed in front of the similar question, should Solution 1 also have an arrow pointing to the student?

---

> ### Author Rebuttal · Authors · 2025-07-31
>
> Thank you for your review. We will address your questions point by point below.
>
> > **1. For example, are the data of the three stages parallel or serial? How are the teaching data of the three stages fused (the size of the dataset before and after the enhancement has not changed, so it can be considered that the data of the three stages have been fused?); Are there no low-quality samples generated? How are they eliminated?**
>
> **Q-1.1: The processing pipeline for the three scenarios' data.**
>
> **A-1.1:** To ensure a fair evaluation of augmentation effects, we used a parallel augmentation pipeline. The original dataset was divided into three equal parts, and each instructional scenario was applied to one part, ensuring one-to-one correspondence between augmented examples and their originals for direct performance comparison.
>
> This design allows us to clearly demonstrate the combined impact of all three instructional scenarios while maintaining sample size consistency and evaluation fairness.
>
> **Q-1.2: Whether the teaching datasets from the three stages were fused, and if so, what specific fusion methodology was applied.**
>
> **A-1.2:** Yes, the augmented dataset integrates samples from three distinct instructional scenarios while maintaining the same total size. To ensure fair evaluation, each augmented sample corresponds one‑to‑one with an original example.
>
> The detailed process is outlined below:
>
> 1. Random Partitioning: The original dataset is split into three disjoint subsets.
>
> 2. Parallel Augmentation: Each subset is processed independently using one of the three teaching scenarios (error correction, debate, analogical reasoning) to generate corresponding augmented examples.
>
> 3. Dataset Construction: The augmented outputs from all three scenarios are shuffled and combined with the remaining original data, ensuring the final augmented dataset has the same total sample count as the original.
>
> This design reflects the key novelty of our approach: rather than merely enlarging the dataset, we enhance its intrinsic quality through rich interactive reasoning structures. As a result, during training, the model more effectively internalizes patterns like "error‑oriented correction," "multi-path reasoning," and "transfer from minimal exemplars."
>
> **Q-1.3: Whether the simulated instructional-interaction process generates any low‑quality samples.**
>
> **A-1.3:** Your concerns are not without merit. Yes, the simulated instructional‑interaction may generate a small number of low‑quality samples. To mitigate this, we implement the following safeguards:
>
> 1. Precise Summarizer Step: The final agent is prompted not only with the interaction history but also with the original dataset’s standard answer. This ensures the final output corrects earlier mistakes and aligns with the original high‑quality answer.
>
> 2. External Verification: We evaluate the augmented data using Qwen2.5‑32B‑Instruct. Only about 4.1% of the enhanced question–answer pairs exhibited procedural errors compared to original outputs.
>
> In addition, results show that BOOST‑QA fine‑tuned models improve significantly more than those trained on the original dataset alone. We plan to add an LLM-based filtering mechanism in the augmentation pipeline to further enhance data quality and robustness.
>
> > **2. Lack of constraints on the entire dialogue process. How to ensure that the dialogue does not deviate from the topic? What constraints are imposed? Is it enough to just define the role?**
>
> **Q-2.1: Constraints to ensure that conversations during the teaching process stay on topic.**
>
> **A-2.1:** By employing the Qwen2.5‑Instruct model family and crafting bespoke, role‑specific system prompts for each agent with clear responsibilities, we ensure that each role strictly adheres to its designated function without interference. This design guarantees that the instructional dialogue remains focused on the topic, preventing role overreach or drifting.
>
> For concrete examples of these prompts, please refer to the appendix in the supplementary materials.  Below is a sample prompt for the teacher agent for your reference:
>
> "You are a teacher responsible for guiding the student’s learning. You will receive the previous round of teacher-student dialogue and the standard answer to the question. Based on the following rules, generate your response:
>
> 1.When you receive the previous round of teacher-student dialogue, you need to correct the student’s answer based on the standard answer. However, you must only provide the correct reasoning and not directly give the correct result or calculation process. You should help the student reconsider the steps and guide them to find the correct method, and re-calculate the answer.
>
> 2.You must avoid providing or hinting at any irrelevant information. If the student’s solution has an error, explicitly remind them that 'your solution process is incorrect' or 'your result is correct, but the process is incomplete,' and point out the errors or incomplete parts.
>
> 3.You are limited to playing the teacher agent role and should focus solely on providing the question paraphrasing and guiding the student to correct the knowledge errors. Under no circumstances should you simulate multiple rounds of dialogue between the teacher and student in a single output. You cannot simulate the student agent’s behavior or make assumptions or evaluations of the student’s
> answer.
>
> 4.You must respond in English."
>
> Under these carefully designed system prompts, each agent clearly understands its role within the simulated classroom dialogue history and performs reasoning that strictly aligns with its assigned responsibilities. This setup prevents agents from overriding one another’s speaking turns or diverging from the topic.
>
> > **3. For example, in the process of Scene 3, if the response to the original question is placed in front of the similar question, should Solution 1 also have an arrow pointing to the student?**
>
> **Q-3.1: The issue where Scene 3 in Flowchart 1 does not align with the description in the main text.**
>
> **A-3.1:** Thank you for your careful reading of our work and your valuable suggestions. Indeed, Figure 1 may currently lead to misunderstanding. To ensure clarity, in the final version of the paper we will revise Figure 1 to explicitly illustrate how the simulated classroom interaction is concatenated in post‑processing, while still clearly presenting the interaction workflow.
>
> > **4. In addition to the issues mentioned in weaknesses, there is another question about the result reporting: Why are the results of ARC and SCI-Q not reported in Table 4?**
>
> **Q-4.1: The issue of missing ARC and SCI-Q benchmarks in Table 4.**
>
> **A-4.1:** Thank you for pointing out this issue. Due to time constraints, the ablation study did not include testing on the ARC and SCI-Q benchmarks. We have added the relevant evaluation experiments and provide the results below:
>
> | Methods | Ori | ME&DB | ME&EP | DB&EP | ME | DB | EP | FULL |
> | :--- | :---: | :---: | :---: | :---: | :---: | :---: | :---: | :---: |
> | **ARC** | 57.76 | 49.57 | 41.13 | 32.34 | 47.35 | 50.26 | 38.14 | **61.52** |
> | **SCI-Q** | 76.50 | 56.30 | 63.60 | 55.20 | 67.90 | 54.40 | 50.20 | **80.10** |
>
> *Table 1: Supplementing Table 4 with the ablation study results.*
>
> ---
> Thank you again for your review and the constructive feedback regarding formatting errors and experimental setup. We will revise the entire manuscript to enhance its professionalism and rigor.

---

> > ### Comment · Reviewer_wU2H · 2025-08-06
> >
> > Thank you for your response, which effectively clarified my concerns about the implementation details, as well as the design of the dialogue process constraints that I had omitted from the appendix. It also supplemented the missing benchmark results report. The addition of relevant content certainly adds clarity and benefit. Therefore, I've raised the score to 4. I hope the new version will incorporate these changes. I also look forward to open-sourcing it.

---

> > > ### Author Response · Authors · 2025-08-06
> > > **Thank you for your feedback and for increasing the score**
> > >
> > > Dear Reviewer wU2H,
> > >
> > > We sincerely thank you for your timely response and your support in raising rating score! We will continue to refine our work based on your suggestions and ensure that the quality of the paper is further improved.
> > >
> > > Best Regards,
> > >
> > > All Authors

---

> ### Author Response · Authors · 2025-08-03
> **We sincerely request your valuable feedback**
>
> Dear Reviewer wU2H,
>
> Thank you very much for your insightful and constructive comments! In our previous detailed response to you, we have addressed your queries comprehensively:
>
> **Q‑1:** We clarified our parallel data augmentation pipeline and how samples from the three instructional scenarios are fused while maintaining dataset size consistency.
>
> **Q‑2:** We described the role‑specific prompts used to enforce dialogue constraints and ensure that agents’ interactions remain focused on the task.
>
> **Q‑3:** We acknowledged the misalignment in Figure 1, committed to revising it to clearly illustrate the multi‑agent classroom interaction workflow.
>
> **Q‑4:** We added the ablation study results for ARC and SCI‑Q to Table 1 (supplement to Table 4 of the paper), as requested.
>
> Moreover, responses from other reviewers may also help you alleviate these concerns. We would be happy to provide further responses otherwise. Look forward to your feedback.
>
> Thanks for your comment again!
>
> Best Regards,
>
> All Authors

---

> ### Comment · Area_Chair_vfTz · 2025-08-06
> **Discussion Period Ending Soon**
>
> Dear Reviewer,
>
> The discussion period is ending soon. We would be grateful if you could take a moment to review the authors' response to your comments and provide any final feedback.
>
> We truly appreciate your time, effort, and valuable contributions to the review process.
>
> Best regards,
>
> AC

---

### Author Response · Authors · 2025-08-09
**Summary of the rebuttal**

Dear ACs/SACs,

We hope this message finds you well. We genuinely appreciate your dedicated efforts in organizing the conference. And we are pleased with the positive feedback from all reviewers. At the end of the discussion phase, we would like to summarize our response to the reviewers.

To address the reviewers' concerns, we have made the following updates and clarification in our response:

---

**Reviewer wU2H**

- The reviewer questioned the parallel/serial nature of three-scenario augmentation, data fusion methods, and quality control (W1); dialogue constraints for topic consistency (W2); Scene 3 flowchart alignment (W3); and missing ARC/SCI-Q results (Q1). We demonstrated parallel execution with 1:1 fused outputs, multi-stage filtering , role-constrained prompting, and post-processing concatenation clarification, while supplementing ARC/SCI-Q results to validate performance gains.

All concerns raised by reviewer wU2H have been addressed through detailed methodological clarifications, targeted prompt design, diagram interpretation, and supplemental experiments, leading to the reviewer’s **score increase from 2 to 4**.

---

**Reviewer sASj**

- The reviewer questioned MASTER's novelty compared to existing role-playing methods (Q1,L1), the authentic implementation of pedagogical principles (Q2), data quality assurance (Q3,L3), and generalization beyond math/programming tasks (L2). We demonstrated MASTER's pedagogical innovation through error-correction, debate and analogical reasoning scenarios; technical rigor via Qwen2.5-Instruct's role-specific prompting; quality control with Precise Summarizer; and broad applicability across general QA benchmarks.

Finally, the reviewer acknowledged that the **concerns have been resolved** and **raised no new concerns**, but we did not receive further feedback. We kindly request the AC to take this issue into account and, if possible, remind the reviewer during the AC-Reviewer discussion stage.

---

**Reviewer vuDb**

- The reviewer inquired about the open-source release of BOOST-QA materials (Q1), computational resources required for data generation (Q2, L2), and potential data leakage concerns regarding performance gains (Q3). We confirmed *code/prompt/dataset release plans*, specified *48 GPU-hours* for 19k sample generation using L20 GPUs, and validated *no data leakage* through embedding drift analysis.

We believe we have thoroughly addressed the concern, but did not receive further feedback. Currently, the reviewer has not provided comments but has submitted the Mandatory Acknowledgement. We kindly request the AC to take this issue into account and, if possible, remind the reviewer during the AC-Reviewer discussion stage.

---

**Reviewer G957**

- The reviewer requested detailed prompt examples and augmented data samples (Q1), comparative analysis with CoT-distillation baselines (Q2), and validation of cross-domain generalization with quality control measures (Q3,L1). We provided complete prompt configurations for all agents, empirical results demonstrating MASTER's superiority over CoT-distillation, and quality verification showing 4.1% error rate via Qwen2.5-32B validation.

We believe we have thoroughly addressed the concern, and the reviewer found the additional experiments in the appendix **helpful in addressing their concerns and had no further questions**.

---

All the concerns raised by all reviewers are addressed by *supplement experiments* and *detailed clarification*.

We would also like to highlight the strengths of our paper recognized by reviewers:

- Introduces novel multi-agent teaching scenarios (**error correction, debate, analogical reasoning**) grounded in educational theory (wU2H, sASj, G957)
- Demonstrates consistent performance gains (5-31 percentage points) across multiple benchmarks (MMLU, CODEMMLU-CC, etc.) (vuDb, wU2H)
- Includes cross-domain generalization validation (G957, vuDb)
- Model-agnostic framework requiring only prompt adjustments for different LLMs (vuDb)
- All reviewers acknowledged that the concerns have been resolved, and Reviewer wU2H is willing to raise the score to 4.0.

---

Thank you again for your effort, dedication, and invaluable support. We appreciate all your help!

Best Regards,
All Authors

---

### Decision · Program_Chairs · 2025-09-17

**Decision:**

Accept (poster)

**Comment:**

### **Main Contribution of the Paper**
- **A Novel Framework for Data Augmentation**: MASTER simulates multi-agent interactions within three pedagogically grounded teaching scenarios: error correction, collaborative debate, and analogical reasoning. This approach addresses the high costs and difficulty of acquiring high-quality fine-tuning data by generating diverse and high-quality teacher-student interaction data.

- **Creation of an Enhanced Fine-tuning Dataset**: By applying the MASTER framework to existing datasets, the authors constructed BOOST-QA. This dataset is specifically designed for fine-tuning LLMs, enriching them with diverse cognitive patterns and enhancing their learnability.

- **Significant Performance Improvement**: Models fine-tuned with the BOOST-QA dataset demonstrate superior performance across multiple benchmarks, showcasing strong multi-task generalization. The MASTER framework is particularly effective in improving the reasoning abilities of models for complex tasks, outperforming models fine-tuned with other baseline methods.


### **Author Response Summarization**
After carefully considering the rebuttal and supplementary materials provided by the authors, I have decided to recommend this paper for acceptance. The authors effectively addressed my initial concerns, particularly those related to implementation details and the constraints of the dialogue process. They have also supplemented the missing benchmark results, which significantly enhanced the clarity and rigor of the study.

The core contribution of this work lies in the novel application of multi-agent simulations, based on pedagogical principles, to augment instruction-tuning data for large language models (LLMs). This framework, called MASTER, generates high-quality reasoning data through three distinct teaching scenarios: error correction, collaborative debate, and analogical reasoning. The resulting BOOST-QA dataset, when used for fine-tuning, has been shown to improve the performance of several base models across multiple benchmarks, particularly in tasks requiring complex reasoning.

The authors' commitment to improving the manuscript and their intention to open-source the code and dataset will be a valuable contribution to the research community.